



# Synergistic effects of previous winter NAO and ENSO on the spring dust activities in North China

**Falei Xu[1], Shuang Wang[1], Yan Li[2], and Juan Feng[1]**

[1]State Key Laboratory of Remote Sensing Science, Faculty of Geographical Science, Beijing Normal University, Beijing, China

[2]Key Laboratory for Semi-Arid Climate Change of the Ministry of Education, College of Atmospheric Sciences, Lanzhou University, Lanzhou, China

**Correspondence:** Juan Feng (fengjuan@bnu.edu.cn)

# Abstract

Dust plays an important role in influencing global weather and climate via impacting the Earth's radiative balance. Based on the atmospheric and oceanic datasets during 1980-2022, the impacts of preceding winter North Atlantic Oscillation (NAO) and El Niño-Southern Oscillation (ENSO) on the following spring dust activities over North China are explored. It is found that both NAO and ENSO exert significant effects in influencing the dust activities over North China, particularly during their negative phases. A synergistic influence on the dust activities in North China is observed when both NAO and ENSO are in negative phase, with their combined impacts exceeding that of either factor alone. The previous winter NAO exhibits significant impacts on the sea surface temperatures (SST) in the North Atlantic, associating with an anomalous SST tripole pattern. Owing to the persistence of SST, these anomalies can extend into the following spring, when anomalous atmospheric teleconnection wave trains would be induced, thereby influencing the dust activities in North China. ENSO, on the one hand, directly impacts dust activities in North China by modulating the circulation in the Western North Pacific (WNP). Moreover, ENSO enhances the NAO's effect on the North Atlantic SST, explaining their synergistic effects on the dust activities over North China. This study explains the combined role of NAO and ENSO on the dust weather over North China, providing one season ahead signals for the forecast of spring dust activities in North China.



## 1. Introduction

Dust, as one of the most significant natural aerosols in the atmosphere, is of great importance to the global radiative balance with its light-absorbing properties, exerting a crucial role in climate change (e.g., Lou et al., 2017; Li et al., 2022; Kok et al., 2023). Moreover, dust not only influences its source regions but also extends its impact across oceans via teleconnections driven by atmospheric circulation. This transboundary transport affects ocean-atmosphere interactions and has a profound impact on the Earth's climate system (Huang et al., 2015). Dust weather, resulting from regional dust surges, poses a formidable threat to socio-economic development, natural ecological environment, as well as human health and safety (e.g., Zhao et al., 2020; Yin et al., 2021; Li et al., 2023). The Gobi Desert in East Asian, especially for the Mongolian Plateau and North China, is a major source of dust (Chen et al., 2023; Hu et al., 2023), contributing approximately 70% of Asia's total dust emissions (Zhang et al., 2003). Given that China is one of the countries most profoundly impacted by dust disasters (Fan et al., 2018), exploring the variations in dust disasters in China is of significant scientific and practical importance.

North China, primarily affected by dust weather, experienced over 80% of its dust events during boreal spring (March-May) (Liu et al., 2022; Shao et al., 2023). In spring, besides the dust source regions over China (mainly Xinjiang and Inner Mongolia), North China also exhibited high dust concentrations and significant dust interannual variability (Liu et al., 2004; Ji and Fan, 2019). Additionally, as a crucial center for politics, economy, and population, it is meaningful to investigate the variations of spring dust weather over North China and to explore the relevant physical mechanisms. Previous studies have revealed that the frequency of dust events in China exhibits strong interannual and interdecadal characteristics, with a high frequency from the 1950s to 1970s, a low frequency from the 1980s to 1990s, and a remarkable increase after 2000 (Zhu et al., 2008; Ji and Fan, 2019). On interdecadal time scales, climate oscillations such as the Atlantic Multidecadal Oscillation (AMO), Pacific Decadal Oscillation (PDO), as well as Antarctic Oscillation (AAO) can influence the dust activities by affecting the climate background. For instance, the positive phase of PDO is favorable for less dust weather by influencing the westerly belt, leading to weaker dust activities (uplift and deposition) in the Asian region (Gong et al., 2006). The AMO plays a role in affecting the global aridification process by altering the thermal properties between land and sea (Huang et al., 2017). Additionally, the AAO may substantially regulate dust weather in China by affecting the frequency of dust in East Asia through the interaction of meridional circulations between the Northern and Southern Hemispheres (Ji and Fan, 2019).



On the interannual scale, a weaker East Asian winter monsoon (EAWM) is associated with
anomalous circulation over the Gobi and Taklamakan deserts facilitate transport of dust,
consequently increasing dust concentrations in China (Lou et al., 2016). The variations of the sea
ice coverage in the Barents Sea can significantly influence the intensity and frequency of dust
weather in China by influencing cyclone generation and thermal instability in North China (Fan et
al., 2018). The North Atlantic Oscillation (NAO) can exert a substantial influence on the spring dust
weather in North China by modulating the zonal wave train from the Atlantic to the Pacific at mid-
latitudes in the Northern Hemisphere, as well as the sea level pressure (SLP) gradient in the Tarim
Basin in China (Zhao et al., 2013). On the synoptic scale, the NAO exerts a vital influence on the
emergence and evolution of dust weather in North China, via its impact on the transport of transient
wave flux and modifications in atmospheric circulation (Li et al., 2023). Beyond extratropical
signals, tropical variabilities, such as El Niño–Southern Oscillation (ENSO), also significantly
modulated dust activities in China by regulating variations in large-scale circulation, precipitation,
and temperature over East Asia (Yang et al., 2022a; Kueh et al., 2023), as well as in Saudi Arabia
(Yu et al., 2015), Central Asia (Xi and Sokolik, 2015), and North America (Achakulwisut et al.,

2017).

From the aforementioned studies on the dust activities in China, it is seen that the NAO and
ENSO are two important factors, with a focus on their individual effects on the dust weather in
China. However, as one of the most significant climate variabilities in the extratropical and tropical
regions, respectively, the NAO and ENSO often co-occur and have complex interactions (López-
Parages et al., 2015). It is found that ENSO can influence the climate near the North Atlantic through
atmospheric forcing of the Pacific North America teleconnection (Wallace and Gutzler, 1981).
During the early winter of El Niño events, strong convective anomalies in the tropical Indian Ocean-
Western Pacific (Abid et al., 2021) and the Gulf of Mexico-Caribbean Sea (Ayarzagüena et al., 2018)
can trigger Rossby wave trains reaching the North Atlantic, leading to positive NAO signals, and
vice versa. Furthermore, the stratosphere, serving as an energy transmission channel, may also be
an important pathway for ENSO to influence the NAO (Jiménez-Esteve and Domeisen, 2018).
Moreover, observations and numerical simulations have demonstrated that NAO signal can induce
a Gill-Matsuno pattern in the tropical region of southern Eurasia, inducing a decadal enhancement
in the linkage between the East Asian summer monsoon (EASM) and ENSO (Wu et al., 2012).
When the NAO is in its positive phase, intensified northeasterlies are observed over tropical North
Atlantic, resulting in increased low-level moisture content and precipitation in the tropical North
Atlantic, paralleling with stronger convection and enhanced ENSO impact (Ding et al., 2023). These





researches highlight the connections and interactions between NAO and ENSO, underscoring the
necessity of considering their synergistic effects on the dust activities in North China.

The synergistic effect refers to the phenomenon where the combined impacts of two or more

factors is significantly greater than their individual role (Li et al., 2019). It is found that there are
synergistic effects in the impact of NAO and ENSO on the weather and climate over China. The
NAO can facilitate the development of the subpolar teleconnection across northern Eurasia
downstream, leading to anomalies in the high-pressure systems over the Ural Mountains and the
Sea of Okhotsk, which in turn affect the EASM (Wang et al., 2000). Meanwhile, ENSO exerts
significant impact on the convective activities in the central Pacific and induces alterations in the
equatorial circulation via the Pacific-East Asia teleconnection, further affecting the atmospheric
circulation and sea surface temperature (SST) in the Western North Pacific (WNP), ultimately
influencing the intensity of EASM (Wang et al., 2000). Therefore, the synergistic effects of these
factors can result in pronounced impacts on the EASM (Wu et al., 2009). During El Niño events,
SST in the central and eastern equatorial Pacific rises, enhancing convective activity near the equator,
which brings more moisture to North China and increases the likelihood of precipitation.
Simultaneously, the positive phase of NAO can alter the atmospheric pressure in the North Atlantic,
influencing the atmospheric circulation over the Eurasian continent. This interaction between NAO
and ENSO synergistically regulates, to some extent, the distribution of precipitation in North China
(Guo et al., 2012).

It is evident that the synergistic effects of NAO and ENSO exert significant impacts on the

climate in China. However, the synergistic impacts of these two factors on the dust events in North
China remains unclear, and the underlying mechanisms and processes are yet to be elucidated.
Therefore, this study will examine the synergistic effects of NAO and ENSO on the dust weather in
North China. Moreover, given that the impacts of winter NAO and ENSO on the climate in China
is more pronounced (Zuo et al., 2016; Zhang et al., 2021b), our analysis will concentrate on the
influence of previous winter NAO and ENSO on the following spring dust, thereby providing a
scientific foundation for predicting dust events in China. The structure of this paper is as follows:
Section 2 outlines the datasets and methods employed in this study. Section 3 presents the analysis
and findings. Section 4 contains the summary and discussion.




## 2. Datasets and methods

### 2.1 Datasets

The dust dataset for the Modern-Era Retrospective Analysis for Research and Applications Version 2 (MERRA-2) was obtained from NASA's Global Modeling and Assimilation Office (GMAO), incorporating assimilated observations from both satellites and ground stations (Gelaro et al., 2017). In this study, the Dust Column Mass Density of the MERRA-2 tavg1_2d_aer_Nx product was utilized to represent the dust concentration with 0.5° × 0.625° resolution. Additionally, the SST dataset was derived from the Hadley Centre of the UK Met Office on a 1°×1° grid (Rayner et al., 2003). The atmospheric reanalysis datasets employed herein were provided from the Fifth Generation Reanalysis Version 5 (ERA5) of the European Centre for Medium-Range Weather Forecasts (ECMWF) with a resolution of 0.25°×0.25° on 37 vertical levels (Hersbach et al., 2020), including wind, geopotential height, and sea-level pressure. Considering the available period of all datasets, the common available period of 1979–2022 was selected. The winter is defined as December-February (December-January-February, DJF), with the winter of 1979 corresponding to the average of December in 1979, January and February in 1980. The spring season is delineated as the average of March-May (March-April-May, MAM).

### 2.2 Methods

The NAO index (NAOI) used is following Li and Wang (2003), quantified by the difference in the normalized monthly SLP regionally zonal averaged over the North Atlantic within 80°W-30°E between 35°N and 65°N. This definition effectively captures the large-scale circulation characteristics associated with NAO, essentially measuring the intensity of zonal winds spanning the entire North Atlantic. Furthermore, ENSO is characterized by Niño3.4 index with SST anomalies averaged over 5°S-5°N, 170°W-120°W (Trenberth, 1997). In this study, we utilized the standardized indices of seasonal averages during 1980-2022, with values exceeding 0.5 standard deviations identified as anomalous years as shown in Table 1.

The memory effect of SST can be elucidated by the SST persistence component ($SST_p$), as delineated in equation (1) (Pan, 2005).

$$SST_p = SST(t) * \frac{Cov[SST(t), SST(t+1)]}{Var[SST(t)]} \quad (1)$$

$SST_p$ represents the memory effect of the previous SST (previous winter) on the following SST (spring), where $SST(t)$ and $SST(t+1)$ denote the previous winter SST and spring SST,



respectively. $Cov[SST(t), SST(t + 1)]$ denotes the covariance between the previous winter SST
and spring SST, while $Var[SST(t)]$ signifies the variance of the previous winter SST.
Consequently, the $Cov[SST(t), SST(t + 1)]/Var[SST(t)]$ represents the connection between the
SST variations in previous winter and spring. A greater value of $SST_p$ indicates the variation of
$SST(t + 1)$ is more closely attached with the variation of $SST(t)$.
The T-N wave activity flux (WAF), formulated by Takaya and Nakamura (2001), represents a
three-dimensional wave action flux that describes the energy dispersion characteristics of stationary
Rossby waves, thereby reflecting the direction of Rossby wave energy dispersion. The WAF is
suitable for application in mid-high latitude regions where the background circulation deviates from
uniform zonality, as obviates the need for the assumption that the basic flow field must be a zonally
averaged basic flow and can accommodate zonally non-uniform wind fields. The convergence and
divergence characteristics of WAF reveal the source and dissipation areas of wave energy, with the
transmission direction being interpretable as the direction of energy transport. The three-
dimensional formulation of WAF is as follows:
$$W = \frac{pcos\varphi}{2|\boldsymbol{U}|} \cdot \begin{pmatrix} \frac{U}{a^2cos^2\varphi}\left[\left(\frac{\partial\psi'}{\partial\lambda}\right)^2 - \psi'\frac{\partial^2\psi'}{\partial\lambda^2}\right] + \frac{V}{a^2cos\varphi}\left[\frac{\partial\psi'}{\partial\lambda}\frac{\partial\psi'}{\partial\varphi} - \psi'\frac{\partial^2\psi'}{\partial\lambda\partial\varphi}\right] \\ \frac{U}{a^2cos\varphi}\left[\frac{\partial\psi'}{\partial\lambda}\frac{\partial\psi'}{\partial\varphi} - \psi'\frac{\partial^2\psi'}{\partial\lambda\partial\varphi}\right] + \frac{V}{a^2}\left[\left(\frac{\partial\psi'}{\partial\varphi}\right)^2 - \psi'\frac{\partial^2\psi'}{\partial\varphi^2}\right] \\ \frac{f_0^2}{N^2}\left\{\frac{U}{acos\varphi}\left[\frac{\partial\psi'}{\partial\lambda}\frac{\partial\psi'}{\partial z} - \psi'\frac{\partial^2\psi'}{\partial\lambda\partial z}\right] + \frac{V}{a}\left[\frac{\partial\psi'}{\partial\varphi}\frac{\partial\psi'}{\partial z} - \psi'\frac{\partial^2\psi'}{\partial\varphi\partial z}\right]\right\} \end{pmatrix} \quad (2)$$

In the expression, $p$, $\varphi$, $\lambda$, $f_0$, and $a$ represent the geopotential height, latitude, longitude,
coriolis parameter, and Earth's radius, respectively. $\psi' = \Phi'/f$ (where $\Phi$ represents the
geopotential) denotes the disturbance of the quasi-geostrophic stream function relative to the
climatology. The basic flow field $\boldsymbol{U} = (U, V)$ denotes the climatic field, where $U$ and $V$ indicate
the zonal and meridional velocities, respectively.
## 3. Results
### 3.1 Impacts of NAO and ENSO on the spring dust in North China
Previous studies have highlighted the significant impacts of NAO (e.g., Wu et al., 2009; Zheng
et al., 2016a; Wang et al., 2018) and ENSO (e.g., Zhao et al., 2016; Zhang et al., 2016; Feng et al.,
2020) on the climate anomalies over China. To investigate their effects on the spring dust, the
correlation between the previous winter NAO and ENSO and following spring dust concentrations
are examined (Figure 1). Significant negative correlations are observed over North China between



NAO and dust content. Similar relationship is seen in the ENSO case. This result indicates a lower
(higher) dust content is expected when NAO and ENSO are in the positive (negative) phases
(Figures 1a-b). Notably, North China is situated at the center of the maximum correlation.
Simultaneously, considering the significant interaction between NAO and ENSO (López-Parages et
al., 2015; Zhang et al., 2015), to detect their independent effects on the dust content, the partial
correlation between NAO (ENSO) and dust content after removing the influence of the ENSO
(NAO) are provided. The results indicate that the significant correlation regions between dust
concentrations and either NAO or ENSO do not change significantly after removing the influence
of the other. These findings suggest a stable and significant connection between NAO and ENSO in
the previous winter and the dust content in North China (Figures 1c-d).

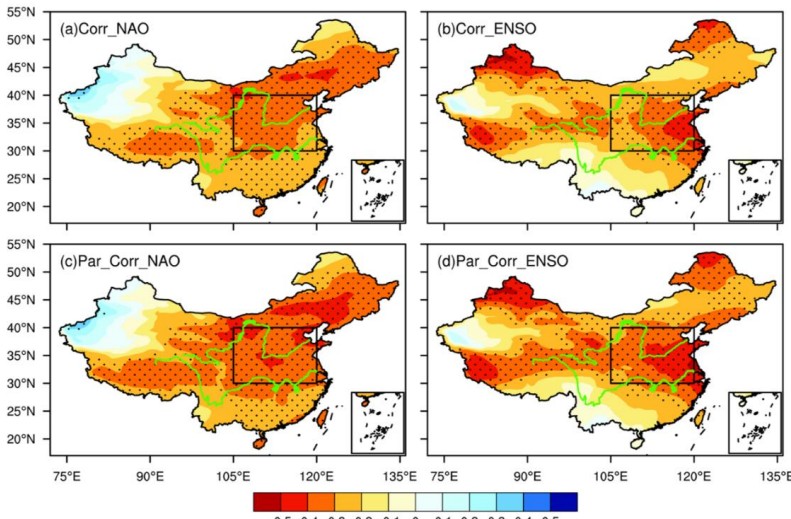


**Figure 1**. (a) Spatial distribution of correlation coefficients between the previous winter NAOI and
spring dust content. (b) As in (a), but with Niño3.4 index. (c) As in (a), but for the partial correlation
after removing the effect of ENSO. (d) As in (c), but after removing the effect of NAO. The black
box represents North China. Stippled areas are statistically significant at the 0.1 level.

Previous studies have indicated that the development rate, intensity variations, and spatial
structure of NAO exhibit distinct asymmetric characteristics between different phases (e.g.,
Feldstein, 2003; Jia et al., 2007). Furthermore, the influence of NAO on the EAWM is more
pronounced during its negative phase (Sung et al., 2010). Similarly, both observational facts and
model experiments suggest that El Niño and La Niña, as the positive and negative phases of ENSO,
are not simply mirror images of each other. The SST anomalies in the tropical Pacific associated



with ENSO exhibit significant asymmetry in terms of meridional range (Zhang et al., 2009),
amplitude (Su et al., 2010), zonal propagation (McPhaden and Zhang, 2009), as well as climate
impact (Feng and Li, 2011; Yang et al.,2022b) under El Niño and La Niña conditions. Consequently,
we further analyzed the connection between NAO/ENSO and spring dust but in different phases.
The results indicate that the relationship between NAO/ENSO and dust in North China also exhibits
significant asymmetry, i.e., with weaker (stronger) correlations during positive (negative) phases of
NAO and ENSO (Figure 2), where significant correlations only appear in the negative phases of
NAO and ENSO. To comprehensively understand the effects of both NAO and ENSO on the dust
activities in North China, the areal average of spring dust content over North China was calculated,
termed as the spring dust index (SDI). Based on the scatter distribution of SDI under different phases
of NAO and ENSO, it is noted that the correlation coefficients between NAOI and SDI during the
positive and negative phases of NAO are -0.46 and -0.05, respectively, indicating that the significant
influence of NAO on the dust in North China mainly occurs during its negative phase (Figure 3a).
Similarly, the correlation distribution between the ENSO and SDI also shows that the influence of
ENSO is more pronounced during its negative phase (Figure 3b). These results indicate that the
impacts of previous winter NAO and ENSO on the spring dust content in North China exhibit
asymmetrical characteristics, significant effects mainly manifested during their negative phases.

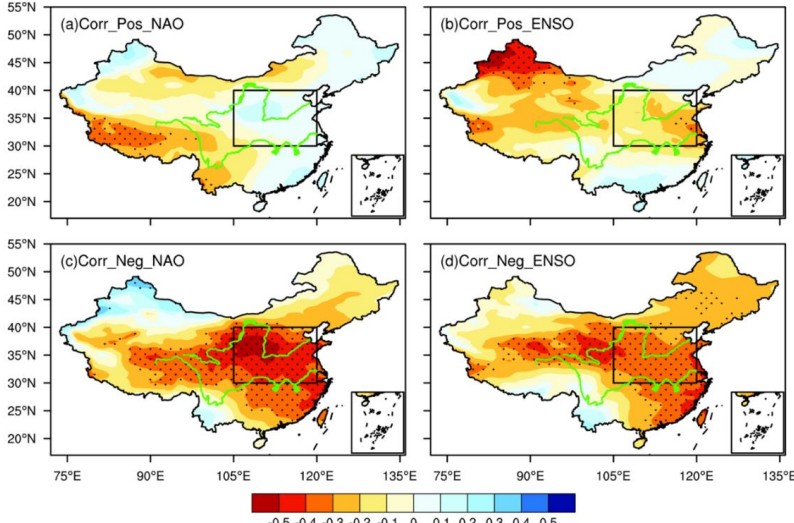


**Figure 2**. Spatial distribution of correlation coefficients between (a) positive and (c) negative NAOI
values and dust content. (b) and (d) As in (a) and (b), respectively, but for the Niño3.4 index.
Stippled areas are statistically significant at the 0.2 level.



The synergistic effects of climate variabilities from mid-high latitudes and tropics are pivotal
mechanisms affecting the weather and climate in East Asia (e.g., Feng et al., 2019; Li et al., 2019).
Correspondingly, we will examine whether the negative phases of previous winter NAO and ENSO
exert synergistic effects on the following spring dust content in North China. As shown in Figure
3c, when the NAO is in its negative phase, including alone occurrence and in conjunction with
negative phase of ENSO, the anomalous values of dust content is 8.32 mg·m$^{-2}$ and 16.21 mg·m$^{-2}$,
respectively. Similarly, the anomalous dust content is 14.88 mg·m$^{-2}$ and 19.40 mg·m$^{-2}$ for the case
of ENSO. When the NAO and ENSO both are in negative phases, the value of dust anomaly (25.23
mg·m$^{-2}$) is much greater than the situation when one of them is in the negative phase. That is the
negative phases of previous winter NAO and ENSO demonstrate synergistic effects on the spring
dust activities in North China. Therefore, three categories, i.e., only the NAO (ENSO) is in its
negative phase, and both NAO and ENSO are in the negative phases (Table 1) are discussed in the
context to elucidate the relevant process of the synergistic effects of NAO and ENSO on the dust
content over North China.

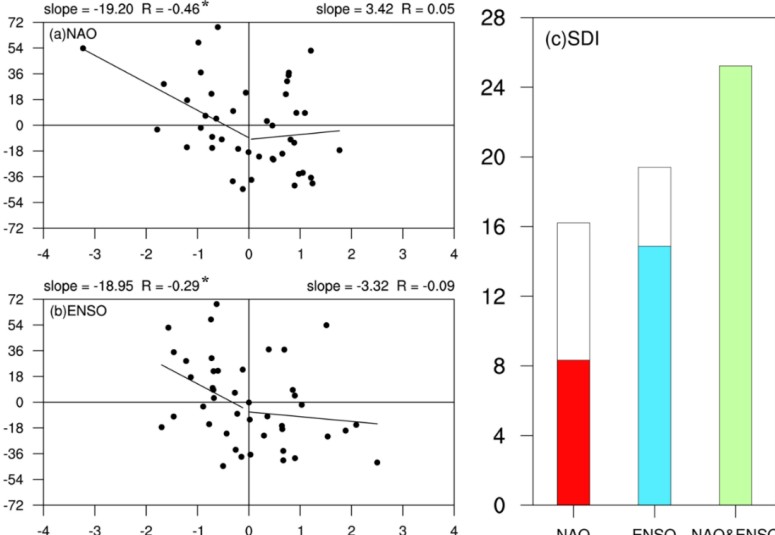


**Figure 3**. Scatterplots of the spring dust content in North China against previous winter (a) NAOI
and (b) Niño3.4 index. Also shown are lines of best fit for positive and negative NAO/Niño3.4 index
values and correlation coefficients (*R*), slope (slope), * indicates significant at the 0.2 level. (c)
Spring dust content over North China during the negative NAO, negative ENSO phases, and
concurrent negative phases of NAO and ENSO (unit: mg·m$^{-2}$). Transparent bars represent negative
phases of the NAO and ENSO, filled bars indicate negative phases of the NAO and ENSO occurring
separately.





**Table 1**. The events of NAO and ENSO classified by three categories during period 1980-2022

| | Years | Numbers |
|---|---|---|
| NAO⁻ | 1980,1982,1985,1986,1987,1996,1998,2001, 2003,2004,2006,2010,2011,2013,2021 | 15 |
| ENSO⁻ | 1984,1985,1986,1989,1996,1999,2000,2001, 2006,2008,2009,2011,2012,2018,2021,2022 | 16 |
| NAO⁻ &ENSO⁻ | 1985,1986,1996,2001,2006,2011,2021 | 7 |

## 3.2 Impacts of NAO and ENSO on the environmental variables

To examine the anomalous characteristics associated with NAO and ENSO, the circulation
anomalies in their solo negative phases, as well as in their co-occur negative phases (Table 1) are
analyzed. In the upper troposphere (200 hPa), the zonal wind is strengthened over the northwest of
China and Mongolia during the negative NAO phase (Figure 4a), with evident positive anomalies
centered around Mongolia, reaching a maximum value of +1.5 m·s⁻¹. In the case of negative ENSO
phase, the upper-level zonal wind also shows an intensification over the northwest region of China
and Mongolia, with a maximum value of +2 m·s⁻¹ (Figure 4d). The intensification of upper-level
zonal wind boosts the upper-level momentum, which is subsequently transferred downward to the
mid-lower troposphere through vertical circulation (Wu et al., 2016; Li et al., 2023), causing windy
weather in the surface dust source regions, facilitating dust lifting and transport activities, thereby
promoting the occurrence of dust weather in the downstream North China. When both the NAO and
ENSO are in their negative phases, the main positive anomaly center appears over North China,
reaching a maximum value of +3 m·s⁻¹, which is stronger than the situation in either the NAO or
ENSO. This result implies the synergistic effects of NAO and ENSO on the upper-level zonal wind,
facilitating an enhanced transport of dust from its source regions to North China, consequently
triggering the onset of dust weather conditions in North China (Figure 4g).
Subsequent analysis delved into the anomalous distribution of the circulation field in the mid
and lower troposphere. In the negative NAO phase, a pronounced 'trough-ridge' anomaly pattern
emerges in the mid-latitude region, characterized by a trough in Siberia and a ridge in the Middle
East, with their anomalous intensities reaching -12 gpm and +10 gpm, respectively (Figure 4b). This
atmospheric configuration fosters a dominant meridional circulation in the mid-high latitude region,
thereby facilitating the enhanced transport of cold air from the north. Such a southward incursion
of cold air serves to strengthen the surface wind speeds, and promote the uplift and transport of dust
from the source regions. In the negative ENSO phase, although the mid-latitude region exhibits a
similar trough-ridge pattern, more pronounced circulation anomalies are observed over the WNP.





At this time, the region is predominantly under the influence of northeasterly winds on its western
flank, manifesting a cyclonic circulation anomaly (Figure 4e), consistent with previous research
results (Ke et al., 2023). This abnormal circulation will hinder the northward transport of warm and
moist air from the South China Sea and the Bay of Bengal, diminishing the likelihood of interactions
with cold air from the north, thus reducing the possibility for the formation of stationary fronts and
precipitation. The decrease in precipitation weakens the wet deposition effect (Zheng et al., 2016b;
Huang et al., 2021), favoring the occurrence of dust weather in the region. When both the NAO and
ENSO are simultaneously in their negative phases, the meridional circulation in the mid-latitude
region is notably enhanced, with the maximum anomalies of the trough and ridge reaching -12 gpm
and +12 gpm, respectively (Figure 4h). Furthermore, the southward shift of the trough-ridge pattern
leads to a more significant increase in wind speed in the upstream dust source regions of North
China, providing a more substantial source of dust for North China. Meanwhile, the presence of a
cyclonic circulation anomaly over the WNP reduces the transport of warm and moist air from the
south, which is unfavorable for precipitation, thereby lowering the wet deposition effect on dust and
further favoring the onset and intensification of dust activities in North China.
As for the SLP, significant positive SLP anomalies appear in Eastern Europe and the Russian
during negative NAO phase, indicative of an intensified Siberian High (SH), which extends
southward to the dust source regions upstream of North China (Figure 4c). The intensification of
the SH typically accompanied with strong northerlies and dry conditions, favoring for the transport
of dust, thereby supplying abundant material sources for dust activities in North China. In the
negative ENSO phase, although the high-latitude region exhibits a weaker SH signal, similar to the
ENSO influence on the circulation pattern in the middle and lower troposphere, more significant
circulation anomalies occur over the WNP. This cyclonic circulation anomaly inhibits the northward
transport of warm and moist air from the south, leading to poorer precipitation conditions in North
China (Figure 4f). When both the NAO and ENSO are in their negative phases, the strength and
influence extent of the SH are more pronounced compared to that when the NAO sole is in negative
phase. Besides, there persists a cyclonic circulation anomaly over the WNP, which is conducive to
the occurrence of dust events in North China (Figure 4i).
The results suggest that when both the NAO and ENSO are in their negative phases, synergistic
effects emerges, rendering the atmospheric circulation in the troposphere more conducive to the
occurrence of dust events in North China. The synergistic effects may be due to the superposition
and interaction of various atmospheric levels and regional characteristics modulated by the NAO



and ENSO, thereby forming more favorable circulation conditions for dust activities in North China.

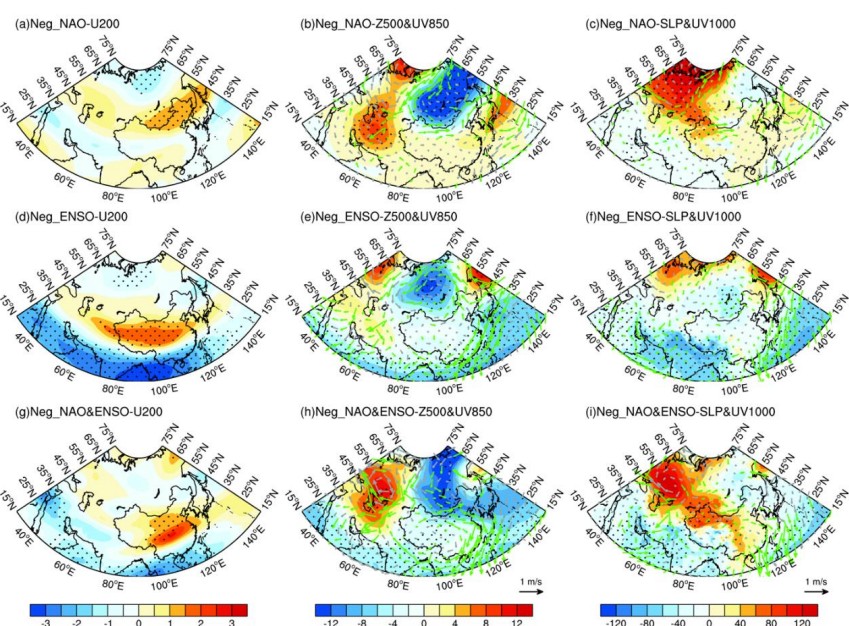


**Figure 4**. Upper, (a) 200 hPa zonal wind anomalies (shading, unit: m·s$^{-1}$), (b) 500 hPa geopotential
height (shading, unit: gpm) and 850 hPa wind field anomalies (arrows, unit: m·s$^{-1}$), (c) sea-level
pressure (shading, unit: Pa) and 1000 hPa wind field anomalies (arrows, unit: m·s$^{-1}$) during the
negative NAO phases. Middle-Lower, as in the upper, but during the negative ENSO phases and
concurrent negative phases of NAO and ENSO, respectively. Stippled areas and green arrows are
statistically significant at the 0.2 level.

Dust activities are multifaceted phenomenon related to large-scale circulation patterns, and

significantly influenced by local surface conditions and meteorological processes. It is found that
surface properties and local meteorological factors play a role in the initiation, development, and
dissipation of dust activities (e.g., Liu et al., 2004; Yao et al., 2021; Huang et al., 2021). In particular,
humidity and precipitation play decisive role in determining the frequency and intensity of dust
activities (Prospero et al., 1987; Kim and Choi, 2015). Low humidity leads to drier soil conditions
in the dust source regions, reducing the cohesion between soil particles and facilitating    dust lifting
and transport activities (Csavina et al., 2014), and vice versa. Similarly, the amount of precipitation
directly affects the wet deposition process of dust. Low precipitation weakens the wet deposition,
resulting in relatively stronger dust activities (Zheng et al., 2016b). Therefore, we further analyzed
their potential impacts on the humidity and precipitation. When the NAO is in its negative phase,
humidity in the spring dust source regions and North China generally reduced, particularly in areas
near the dust source regions, indicating that these areas are conducive to dust transport and prone to





causing dust weather in North China (Figure 5a). As for the precipitation, there is more spring
precipitation in the northwest region of China, while precipitation in the Mongolia and the North
China is relatively less (Figure 5b). In the negative ENSO phase, the variation in humidity is similar
to that during the negative NAO phase, but with a greater amplitude (Figure 5c), indicating that
ENSO has a stronger impact on the humidity conditions in North China. Moreover, the precipitation
shows a significant abnormal decrease over Mongolia and North China, which is highly conducive
to dust activities and the generation of dust weather (Figure 5d). When both the NAO and ENSO
are in the negative phases, the humidity anomalies in the dust source regions and North China are
more intense than the individual factor (Figure 5e). The variation in precipitation are similar to those
in humidity, the reduction in precipitation in the dust source regions and North China exceeds the
sole role (Figure 5f). The aforementioned analysis indicates that NAO and ENSO can modulate
humidity and precipitation, ultimately affecting dust weather. During the negative NAO phase, the
diminished atmospheric pressure gradient in the mid-high latitude regions of North Atlantic leads
to the intensification and southward shift of the SH (Zhou et al., 2023), accompanied by strong wind,
making drier and conducive to dust lifting and transport in the dust source regions. In the negative
ENSO phase, the upper atmosphere over the WNP is dominated by significant negative anomalies
in geopotential height and northeasterly winds (Zhang et al., 2015), reducing moist transport. When
the NAO and ENSO both are in negative phases, their regulation of atmospheric circulation
produces synergistic effects, further influencing the variations of humidity and precipitation, thereby
promoting the occurrence and development of dust activities in North China.



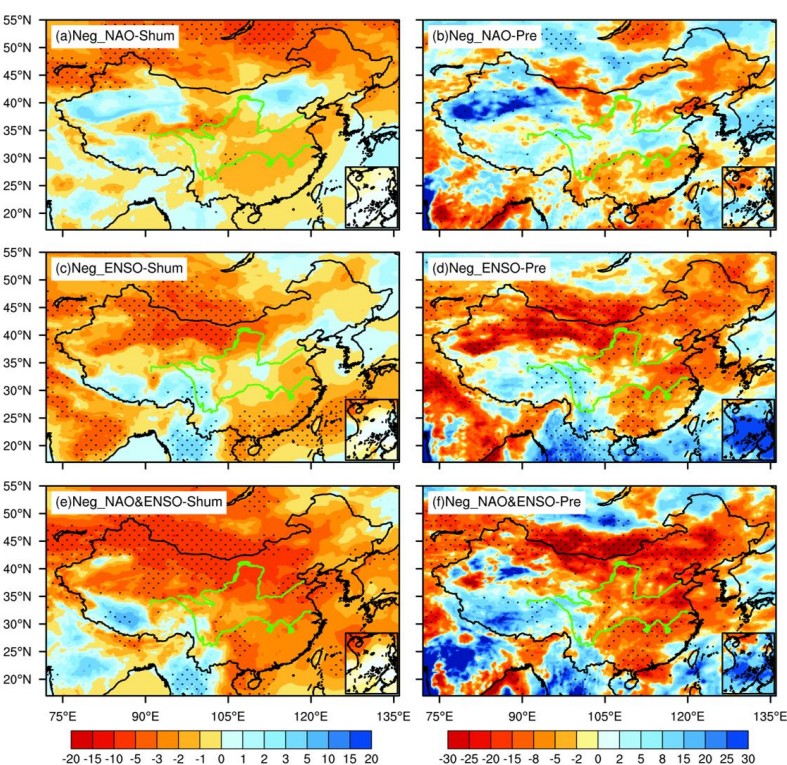

**Figure 5**. Upper, composite percentage anomalies of (a) humidity and (b) precipitation during negative NAO phases. Middle-Lower, as in the upper, but during negative ENSO phases and concurrent negative phases of NAO and ENSO, respectively. Stippled areas are statistically significant at the 0.2 level.

### 3.3 Physical Mechanisms of the NAO and ENSO on the dust weather

The above results demonstrated that the previous winter NAO and ENSO exert significant impacts on the spring dust activities in North China. Consequently, an examination of the underlying physical mechanisms is warranted. Given the relatively short memory of NAO as an atmospheric phenomenon, we will employ the concept of ocean-atmosphere coupling bridge to elucidate the involved processes. The previous ENSO signal can alter the atmospheric circulation over the WNP through the persistent impact of SST, thereby significantly affecting subsequent weather and climate in China (e.g., Wu et al., 2017; Kim and Kug, 2018; Jiang et al., 2019). The tripole configuration of SST is the leading mode of SST variation in the North Atlantic, and its variabilities are closely associated with the NAO (Czaja and Frankignoul, 2002; Wu et al., 2009; Figure 7a), which allows the previous NAO signal to exert a long-term influence on the subsequent weather and climate in




China (e.g., Wu et al., 2012; Zhang et al., 2021a; Li et al., 2023). The variation of SDI is linked with
an anomalous tripole SST in the North Atlantic (Figure 6a), paralleling with the SST anomalies
accompanied with the negative phase of NAO. Therefore, the North Atlantic tripole index (NATI)
is further delineated (Equations 3-6), as well as the relationships among the NAOI, NATI, and SDI
are explored. The correlation analysis between the high and low years of SDI and NATI reveals a
pronounced difference, indicating an asymmetric correlation (Figures 6b-c). Specifically, the
significant relationship between SDI and NATI only existed in the positive SDI years, implying the
occurrence of NATI would connected with more dust weather over North China.

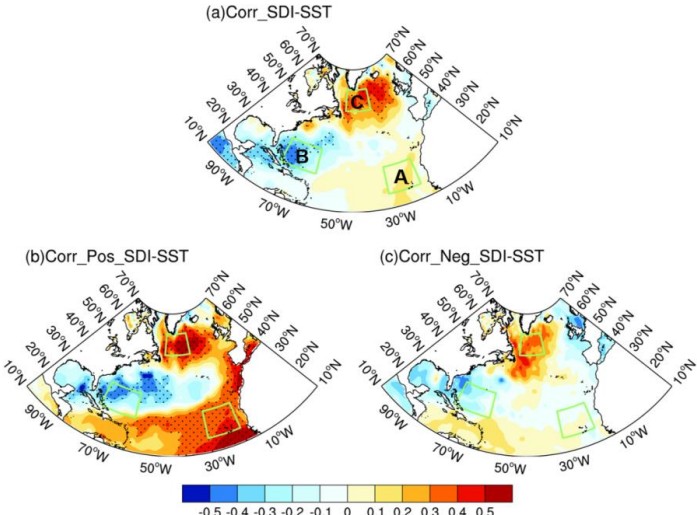


**Figure 6**. (a) Spatial distribution of the correlation coefficients between the spring SDI and
simultaneous SST. (b)-(c) As in (a), but for the positive and negative phase of SDI. Stippled areas
are statistically significant at the 0.2 level.
$$SST_A = [15-25°N, 32-20°W] \qquad (3)$$

$$SST_B = [22-32°N, 75-60°W] \qquad (4)$$

$$SST_C = [50-60°N, 50-32°W] \qquad (5)$$

$$NATI = SST_B - \frac{1}{2}(SST_A + SST_C) \qquad (6)$$

Subsequent analyses delved into the association between the previous winter NAO and the
North Atlantic SST. It is seen that the correlation coefficients between the negative (positive) NAOI
and NATI are 0.41(-0.09) (figures not shown), indicating that the influence of previous winter NAO
on the following spring NATI only manifest during its negative phase. This elucidates the reason
why the significant impact of NAO on the dust activities in North China only existed during its
negative phase. In the negative NAO phase, there is a notable correlation between the previous





winter NATI and the spring SST and $SST_p$ (Figures 7b-c), indicating that the previous winter NATI
can persist to spring, in which the self-persistence of SST playing a crucial role. Similar findings
are observed during the negative phase of ENSO (Figures 7d-f) and when both the NAO and ENSO
occur simultaneously (Figures 7g-i).
The correlation between the previous winter NAO and North Atlantic SST reveals that in the
NAO negative phase (Figure 7a), the variation of NAO is linked with an anomalous tripole SST
pattern in the North Atlantic. Meanwhile, similar findings are observed when negative ENSO events
occur (Figure 7d). This suggests that there may be a positive feedback occurred between NAO and
North Atlantic SST during negative ENSO phase. When both the NAO and ENSO are in the
negative phases, the anomalous tripole SST pattern is more pronounced (Figure 7g). This further
elucidates that ENSO exerts a promoting effect on strengthening the connection between the
negative NAO and NATI, thereby providing an explanation for the synergistic effects of the NAO
and ENSO on the dust weather in North China. Additionally, the correlation coefficients between
the NAOI and NATI under different scenarios can illustrate the synergistic influence of the NAO
and ENSO on the persistence of SST anomalies (Table 2). Specifically, when the negative phase of
NAO and ENSO occur together, the correlation coefficients between the NAOI and NATI are greater
than those influenced by a single factor alone (Table 2).

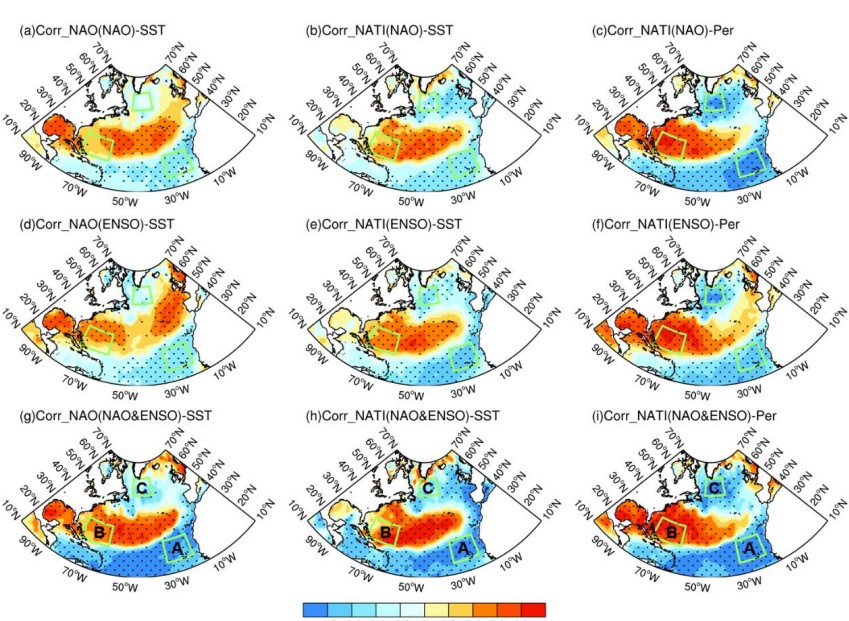


**Figure 7**. Upper, correlation distributions of the (a) winter NAOI with winter SST, (b) winter NATI





with spring SST, and (c) winter NATI with $SST_p$ during negative NAO phases. Middle-Lower, as
in the upper, but during the negative ENSO phases and concurrent negative phases of NAO and
ENSO, respectively. Stippled areas are statistically significant at the 0.2 level.
**Table 2**. Correlation coefficients between the NAOI and NATI in three different categories. *
indicates significant at the 0.1 level.

|  | DJF_NAO & DJF _NATI | DJF_NATI & MAM_NATI |
|---|---|---|
| NAO- phase | 0.41* | 0.51* |
| ENSO- phase | 0.52* | 0.69* |
| NAO- & ENSO- phase | 0.66* | 0.69* |

The NAO preserves its anomalous signal within the tripole SST during the previous winter,
and releases the signal in the following spring. Given the distance across the entire Eurasian
continent between the North Atlantic and North China, the role of teleconnection wave trains is
particularly important in influencing dust activities over North China. Figure 8a illustrates the
geopotential height field at 200 hPa regressed onto the spring NATI during the negative phase of
NAO. This reveals a pronounced north-south reversed dipole pattern in the North Atlantic, i.e.,
negative over Azores and positive over Iceland, representing a typical negative NAO structure (e.g.,
Wallace and Gutzler, 1981; Hurrell, 1995; Li and Wang, 2003). Meanwhile, a positive-negative-
positive teleconnection wave train structure centered around eastern Europe, Middle East, and North
China is observed, suggesting that the disturbance energy propagates downstream from the North
Atlantic through waveguide effects, leading to an anticyclonic circulation anomaly in North China.
Similar teleconnection wave-train propagation characteristics are also observed in the 200 hPa
meridional wind and vorticity fields (Figure 8b, c). During the negative phase of ENSO, modulated
by the NATI, analogous teleconnection structures are also seen in the circulation field (Figure 8d-
f). Notably, when the NAO and ENSO are both in their negative phases, the teleconnection structure
reflected in the circulation field is more pronounced than when only one factor is dominated (Figure
8g-i), confirming the synergistic effects of both factors on the circulation processes affecting dust
activities in North China.



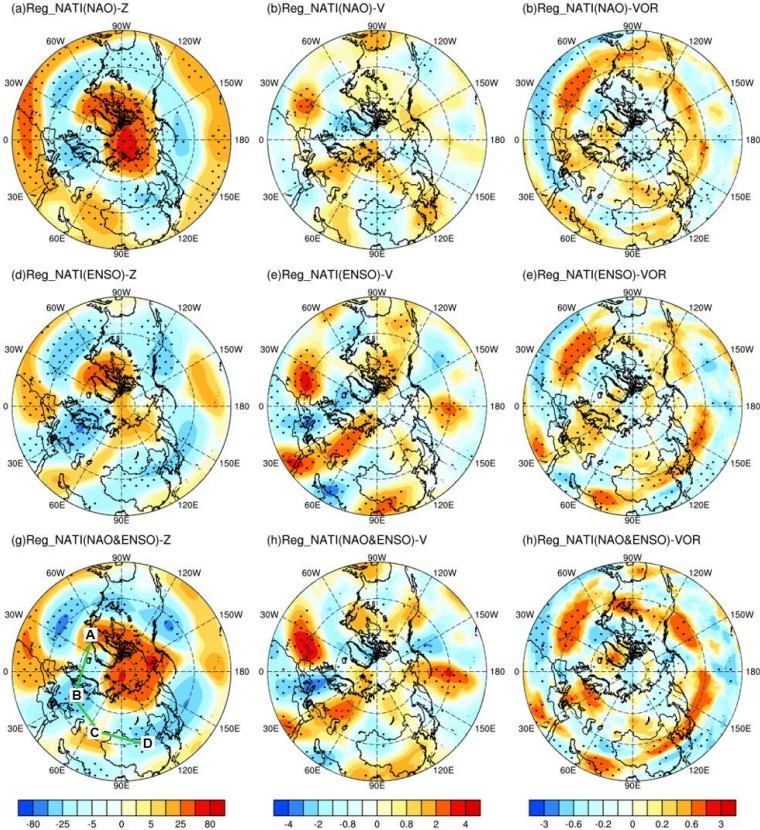


**Figure 8**. Upper, regression distribution of spring NATI against the spring (a) geopotential height
(unit: gpm), (b) meridional wind (unit: m·s$^{-1}$), and (c) vorticity (unit: 10$^{-5}$·m·s$^{-1}$) at 200 hPa during
the negative NAO phase. Middle-lower, as in the upper, but during the negative ENSO phases and
concurrent negative phases of NAO and ENSO, respectively. Regression fields multiplied by -1.
Stippled areas are statistically significant at the 0.2 level.

In order to further examine the impact mechanisms of the NAO and ENSO on the spring dust
activities in North China, based on the propagation characteristics of the teleconnection wave train
shown in Figure 8, the distribution of cross-section of the geopotential height field is presented
(Figure 9). When both the NAO and ENSO are in their negative phases, the NATI anomalies
correspond to the teleconnection wave train extending from the upper to lower troposphere, which
is specifically characterized by a positive-negative-positive tripole pattern. This wave train
propagates from the North Atlantic, traversing eastern Europe and Middle East, and ultimately
influencing circulation processes associated with the dust weather over North China. Furthermore,
the analysis of cross-section at different levels of the troposphere reveals that under the negative




phases of NAO and ENSO, the teleconnection wave train excited by the NATI exhibits quasi-
barotropic features, with this anomalous structure being primarily concentrated in the middle-upper
troposphere. When the NAO and ENSO are simultaneously in their negative phases, the intensity
and scope of the teleconnection wave train are significantly enhanced and expanded compared to
the influence of a single factor (Figure 9c), demonstrating synergistic effects.

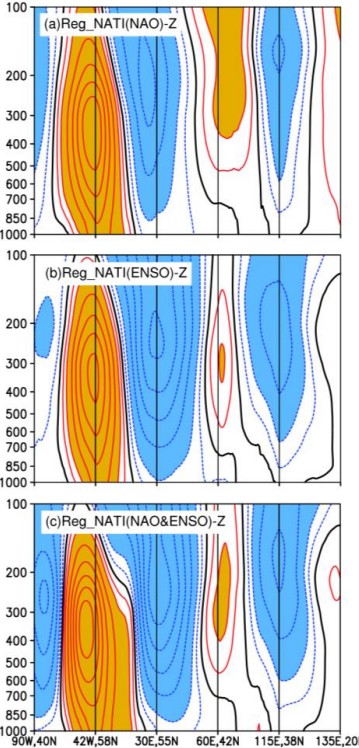


**Figure 9**. Vertical section of regression of spring NATI against the geopotential height along the
solid line labeled A (42°W, 58°N), B (30°E, 55°N), C (60°E, 42°N), and D (115°E, 38°N) in Figure
8g for (a) negative NAO phase in the previous winter. Panels (b)-(c) as in (a), but during the negative
ENSO phases and concurrent negative phases of NAO and ENSO, respectively (unit: gpm).
Regression fields have multiplied by -1. Shading indicates the absolute value is greater than 10 gpm.
To provide a more comprehensive analysis of the transport process of disturbance energy in
the atmosphere, the horizontal distribution of the WAF associated with spring NATI variations is
further examined. Under the scenario that either the NAO or ENSO is in their negative phases, WAF
can be clearly observed to originate from the North Atlantic, traverse the Eurasian continent, and
extend to the North China (Figures 10a-b). When both factors occur simultaneously, not only is the
transport intensity of WAF enhanced, but its impact range on the dust weather in North China is also



broadened (Figure 10c). Through the analysis of teleconnection wave trains and WAF, it is
determined that the synergistic effects not only enhance the disturbance intensity in the atmosphere
but also expand impact range, thereby promoting the occurrence and development of spring dust
weather in North China. The enhancement and expansion of atmospheric disturbances may be
related to large-scale circulation anomalies and local climate condition changes induced by the
synergistic effects of the NAO and ENSO, which in turn affect the transport and deposition
processes of dust.

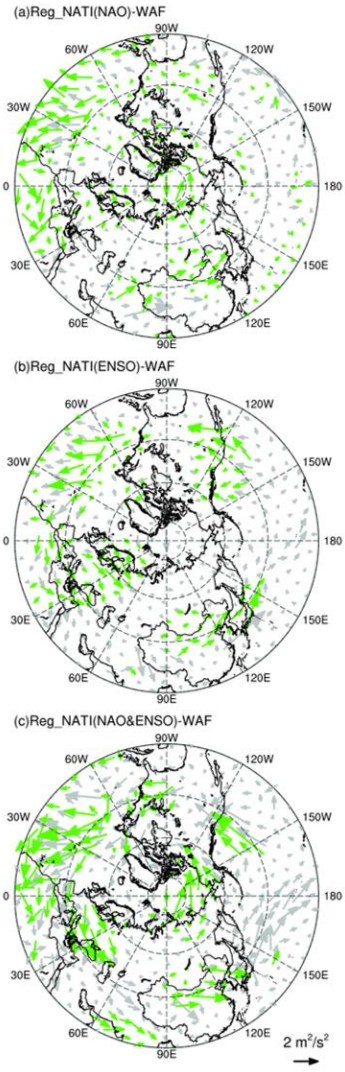


**Figure 10**. Upper, regression distribution of spring NATI against the T-N wave activity flux (a)
during negative NAO phase. Middle-lower, as in upper, but during the negative ENSO phases and



concurrent negative phases of NAO and ENSO, respectively (units: m²·s⁻²). Regression fields have
multiplied by -1. Green arrows are statistically significant at the 0.2 level.

## 4. Conclusions and discussions

The NAO and ENSO exert significant impacts on climate variability in China (e.g., Zhang et
al., 2016; Wang et al., 2018; Feng et al., 2020). Although North China is not the primary dust source,
dusty disasters are notably active in this region during spring. This study highlights that the previous
winter NAO and ENSO exert essential influences on the following spring dust activities in North
China. Their impacts are asymmetric, manifesting only when both are in their negative phases.
Furthermore, the results indicate that NAO and ENSO in the negative phase have synergistic effects
on the spring dust activities in North China, promoting dust activities and with greater impacts than
their sole effect.
Under the regulatory influence of the negative phases of NAO and ENSO, the atmospheric
circulation in the troposphere from the lower to upper layers exhibits anomalies, including variations
in the upper-level zonal winds, mid-latitude trough-ridge systems, circulation over the WNP, and
SH at the SLP. These variations promote the occurrence and development of dust weather in North
China. Simultaneously, accompanying anomalies in the atmospheric circulation pattern also affect
local meteorological factors, including humidity and precipitation, which in turn show impacts on
the dust activities in North China. Notably, when both the NAO and ENSO are in their negative
phases, synergistic effects occur, making the anomalies in atmospheric circulation from the lower
to upper layers, as well as variations in humidity and precipitation, more conducive to the occurrence
of dust events in North China. The impact of NAO on the underlying SST pattern is predominantly
observed during its negative phase, elucidating why the NAO significantly influences dust activities
in North China only during its negative phase. Furthermore, when both the NAO and ENSO
simultaneously manifest in their negative phases, the teleconnection wave trains and WAF
stimulated from the North Atlantic are more intense, thereby more effectively influencing dust
activities in North China, indicating the synergistic effects of the two variabilities on the dust
activities over North China.
In the process where the previous winter NAO and ENSO affect the following spring dust
activities in North China, the persistence of anomalous NAT over North Atlantic plays an important
role. The previous winter NAO stores its signal in the NAT (Czaja and Frankignoul, 2002; Wu et
al., 2009). Due to the persistence of SST, the anomalous NAT can last from winter to spring (e.g.,



Wu et al., 2012; Zhang et al., 2021a; Li et al., 2023). In spring, NAT regulates the circulation pattern
in North China through teleconnection wave trains, ultimately affecting the dust activities over
North China. The signal of previous winter ENSO can persist into spring, due to the persistence of
SST, and it affects the dust activities in North China through two pathways: i.e., directly influencing
the dust activities in North China by affecting the circulation anomalies over the WNP, and playing
a facilitating role in the process where the NAO excites NAT, thereby affecting the dust activities in
North China. This provides a plausible explanation why the previous winter NAO and ENSO exert
synergistic effects on the following spring dust activities in North China.
This study investigated the impacts of NAO and ENSO on the dust activities in North China
and the involved physical processes, indicating the one season ahead signals provide as the useful
predictors for the spring dust activities in North China. Future work will focus on developing a
forecast model using the NAO and ENSO as predictors and validating its prediction effectiveness.
Additionally, as previous studies have highlighted strong interdecadal variations are existed in both
NAO and ENSO (Woollings et al., 2015; Dieppois et al., 2021; Wang et al., 2023), it is of interest
to further detect whether the synergistic effects of NAO and ENSO on the dusty activity over North
China experience interdecadal variations. However, due to the availability of dataset, the potential
impacts of the interdecadal variability of the NAO and ENSO on dust activities have not been
discussed in this study. Simultaneously, as reported that the state-of-art models can reproduce the
individual impact of NAO and ENSO on the dust activities in North China (Ginoux et al., 2004;
Yang et al., 2022a), whether their synergistic effects on the dust weather could be well simulated,
requiring further researches. Additionally, previous studies have indicated that the variability of
ENSO is likely to intensify under the background of global warming (Cai et al., 2021). Therefore,
it is crucial to investigate the future changes in the NAO, as well as future change of its synergistic
effects with the ENSO on the dust weather, to better understand the plausible trends of future dust
activities in North China.

**Code and data availability.** The MERRA-2 dust aerosol concentrations dataset can be downloaded
from https://disc.gsfc.nasa.gov/datasets?project=MERRA-2 (last access: 28 March 2024). The
atmospheric reanalysis datasets, including the wind field, geopotential height field, and sea level
pressure        field        can        be        downloaded        from
https://cds.climate.copernicus.eu/#!/search?text=ERA5&type=dataset (last access: 28 March 2024).
Our results can be made available upon request. The oceanic reanalysis data can be downloaded



from https://www.metoffice.gov.uk/hadobs/hadisst (last access: 28 March 2024). Our results can be made available upon request.

**Author contributions.** FLX and JF conceptualized and designed the research. FLX and JF synthesized and analyzed the data. FLX, SW, YL, and JF produced the figures. FLX and SW contributed to the datasets retrieval. All the authors discussed the results and wrote the paper.

**Competing interests.** The authors declare that they have no conflict of interest.

**Disclaimer.** Publisher's note: Copernicus Publications remains neutral with regard to jurisdictional claims in published maps and institutional affiliations.

**Acknowledgements.** This work was jointly supported by the National Key Research and Development Program of China (2023YFF0805100), the BNU-FGS Global Environmental Change Program (No. 2023-GC-ZYTS-03), and the State Key Laboratory of Tropical Oceanography, South China Sea Institute of Oceanology, Chinese Academy of Sciences (Project No. LTO2310).

**Financial support.** This work was jointly supported by the National Key Research and Development Program of China (2023YFF0805100), the BNU-FGS Global Environmental Change Program (No. 2023-GC-ZYTS-03), and the State Key Laboratory of Tropical Oceanography, South China Sea Institute of Oceanology, Chinese Academy of Sciences (Project No. LTO2310).





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
