# Peer review of "Synergistic effects of previous winter NAO and ENSO on the spring dust activities in North China"

_EGUsphere, 2024_

## Referee Comment (RC1)

*General comments*

*This study investigated the impacts of preceding boreal winter North Atlantic Oscillation (NAO) and El Niño-Southern Oscillation (ENSO) on the following spring dust activities over North China during 1980-2022. The authors demonstrated that the significant impacts of NAO and ENSO on the dust activities over North China is only manifested in the negative phases, and discussed the physical mechanism involved to illustrate why the negative phases of NAO and ENSO show a synergistic effect on the following dust events in North China. The message is conveyed clearly and the topic is interesting. The results of this study provide an insight to further understand the dust activities over North China. The conclusions are substantiated based on composite analyses. If published, this work could serve as a valuable reference for dust weather. However, it needs to be minor revised before accepted this paper for publication in ACP with addressing those comments listed below:*

**Specific comments are as follows:**

*1. The NAO is a large-scale seesaw in atmospheric mass between the subtropical high and the polar low. It is the dominant mode of atmospheric circulation variability in the North Atlantic sector throughout the year. The definition of the NAO index derived using EOF is commonly employed to depict the variation of NAO. However, the SLP difference between 35°N and 65°N within the Atlantic section is used to define the NAO index. A full comparison of the NAO index is necessary to establish the robustness of result.*

*2. The authors focus on the relationship between preceding winter NAO and ENSO and dust weather in late spring. The introduction mentions that "the impacts of winter NAO and ENSO on the climate in China is more pronounced" by citing results from previous work. However, it is unclear whether the cross-seasonal impacts also apply when exploring the relationship between NAO, ENSO and dust weather in North China. Therefore, it would be better to provide some references to explain why we should investigate the impacts of previous winter of NAO and*

*ENSO on spring dust weather.*

*3. In the paper, the authors primarily discuss the effect of NAO and ENSO negative phases on the dust activities over North China. However, given the various phases combinations between these two factors, a more detailed explanation as to why only the negative-negative combinations are considered.*

*4. In Figures 1-2, the authors illustrate the relationship between NAO, ENSO and dust weather over North China through the spatial distribution of correlation coefficients, and that the relationship is only manifested when NAO and ENSO are in negative phases. A quantitative analysis is needed to further establish the robustness of the result.*

*5. From Fig 3 and Table 1, it is evident that there are two types when NAO and ENSO are in their negative phases: negative phases of the NAO and ENSO, and negative phases of the NAO and ENSO occurring separately (remove the years with concurrent negative phases of NAO and ENSO). Furthermore, the subsequent composite analyses in the study, focus on the cases with negative phases of the NAO and ENSO. The authors should explain why they have made this choice.*

*6. In Figs 4 c, f, and i, the variations in the near-surface wind field caused by anomalies of Siberian High, lead to dust emissions from the source areas. However, the depiction of the wind field anomalies appears unclear. It is recommended to modify the Figs to highlight the variations in the wind field.*

*7. In Table 2, the value of correlation coefficients between the previous winter NATI and spring NATI are similar in scenarios of ENSO- phase (when the negative phase of ENSO occurs alone) and NAO- & ENSO- phase (when the negative phases of both NAO and ENSO co-occur). However, if there exists a synergistic effect of NAO and ENSO on the dust weather, the correlation in the scenario where both NAO and*

*ENSO negative phases co-occur should be higher than when the negative phases of NAO and ENSO occur separately. The authors should have provided a more detailed explanation to clarify this point.*

*8. The main mechanism for the impact of the winter NAO on the spring dust is the maintenance of the North Atlantic SST anomalies from winter to spring, consistent with previous findings (Chen et al. 2020; Wu and Chen 2020; Song et al. 2022). Several discussions could be added.*

*Song, L.-Y., et al, 2022: Distinct evolutions of haze pollution from winter to following spring over the North China Plain: Role of the North Atlantic sea surface temperature anomalies. Atmos. Chem. Phys., 22, 1669–1688.*

*Wu and Chen, 2020: What leads to persisting surface air temperature anomalies from winter to following spring over the mid-high latitude Eurasia?. Journal of Climate, 33, 5861-5883.*

*Chen et al. 2020: Strengthened connection between springtime North Atlantic Oscillation and North Atlantic tripole SST pattern since the late-1980s. Journal of Climate, 35(5), 2007-2022.*

*9. There are lots of clerical errors, i.e.,*

 *Line 17-18, sea surface temperatures (SST) in the North Atlantic*

 *Line 220, with regard to the description of the graphs, there may be some errors that*

 *"(b) and (d) As in (a) and (b) " -> "(c) and (d) As in (a) and (b)".*

 *The authors should carefully check the whole manuscript.*

---

## Author Comment (AC1)

**Response to Comments of Reviewer 1**

**Manuscript number**: egusphere-2024-955

**Author(s)**: Falei Xu, Shuang Wang, Yan Li, and Juan Feng

**Title**: Synergistic effects of previous winter NAO and ENSO on the spring dust activities in North China

**General comments:**

This study investigated the impacts of preceding boreal winter North Atlantic Oscillation (NAO) and El Niño-Southern Oscillation (ENSO) on the following spring dust activities over North China during 1980-2022. The authors demonstrated that the significant impacts of NAO and ENSO on the dust activities over North China is only manifested in the negative phases, and discussed the physical mechanism involved to illustrate why the negative phases of NAO and ENSO show a synergistic effect on the following dust events in North China. The message is conveyed clearly and the topic is interesting. The results of this study provide an insight to further understand the dust activities over North China. The conclusions are substantiated based on composite analyses. If published, this work could serve as a valuable reference for dust weather. However, it needs to be minor revised before accepted this paper for publication in ACP with addressing those comments listed below:

**Response:**

Thanks to the reviewer for the helpful comments and suggestions. We have revised the manuscript seriously and carefully according to the reviewer's comments and suggestions. The point-to-point responses to the comments are listed as follows.

**Specific comments are as follows:**

1. The NAO is a large-scale seesaw in atmospheric mass between the subtropical high and the polar low. It is the dominant mode of atmospheric circulation variability in the North Atlantic sector throughout the year. The definition of the NAO index derived using EOF is commonly employed to depict the variation of NAO. However, the SLP difference between 35°N and 65°N within the Atlantic section is used to define the NAO index. A full comparison of the NAO index is necessary to establish the robustness of result.

**Response:**

Thank you for the comments.

The NAO index (NAOI) employed in the manuscript, is defined as the differences of normalized sea level pressures regionally zonal-averaged over the North Atlantic sector (Li and Wang, 2003). The NAO index captures well large-scale circulation features of the NAO, and is essentially a measure of the intensity of zonal winds across the central North Atlantic between 35°N to 65°N. A systematic comparison of six NAO indices (Rogers, 1984; Barnston and Livezey, 1987; Moses et al., 1987; Hurrell, 1995; Jones et al.,1997; Li and Wang, 2003), shows that the NAOI employed in the manuscript provides a much more faithful and optimal representation of the spatial-temporal variability associated with the NAO, suggesting the NAOI maybe as a suitable choice for describing and monitoring variability of the broad-scale NAO and for diagnosing relationships between the NAO and global climate variations (Li and Wang, 2003).

We also employ the NAOI produce by Hurrell (1995) and Jones (1997), which have been used in many studies (e.g., Wang et al.,2022; Najibi et al., 2023; Parry et al., 2023), for correlation analysis with the NAOI used in this manuscript. A good agreement with correlation coefficients of 0.96 and 0.94 between these two indices and the NAOI used in this manuscript (Figure R1). As well as, using the NAOI provided by Hurrell (1995) and Jones (1997), the asymmetric impact and the synergistic effects with ENSO on dust activities over North China of NAOI still remain (Figures R2-R4). Therefore, the robustness of the results will not be affected by the NAOI verified by

above process. We have added the description into the revised manuscript, as "And we use the NAOI provided by Hurrell (1995) and Jones (1997), which have been used in many studies (e.g., Wang et al.,2022; Najibi et al., 2023; Parry et al., 2023), for correlation analysis with the NAOI used in this work and find a good agreement with a correlation coefficient of 0.96 and 0.94. As well as, using the NAOI provided by Hurrell (1995) and Jones (1997), the asymmetric impact and the synergistic effects with ENSO on dust activities over North China of NAO still remain (figure not shown). This point indicates that the result would not be affected by choice of NAOI", as shown in Lines 152-154.

[Figure]

**Figure R1**. (a) The winter NAOI used in the manuscript (black line) and provided by Hurrell (blue line) and Jones (red line) during 1980-2022.

[Figure]

**Figure R2**. Spatial distribution of correlation coefficients between the previous winter NAOI and spring dust content (a). (b-c) As in (a), but for the NAOI produce by Jones and Hurrell, respectively. (d-f) As in (a-c), but for the partial correlation after removing the effect of ENSO. The black box represents North China. Stippled areas are statistically significant at the 0.1 level.

[Figure]

**Figure R3**. Spatial distribution of correlation coefficients between (a) positive and (b) negative NAOI values and dust content. (c-d) and (e-f), As in (a-b), but for the NAOI produce by Jones and Hurrell, respectively. The black box represents North China. Stippled areas are statistically significant at the 0.2 level.

[Figure]

**Figure R4**. Spring dust content over North China during the negative NAO, negative ENSO phases, and concurrent negative phases of NAO and ENSO. Transparent bars represent negative phases of

the NAO and ENSO, filled bars indicate negative phases of the NAO and ENSO occurring separately and co- occurring (a). (b-c) As in (a), but for the NAOI produce by Jones and Hurrell, respectively (unit: mg m$^{-2}$).

2. The authors focus on the relationship between preceding winter NAO and ENSO and dust weather in late spring. The introduction mentions that "the impacts of winter NAO and ENSO on the climate in China is more pronounced" by citing results from previous work. However, it is unclear whether the cross-seasonal impacts also apply when exploring the relationship between NAO, ENSO and dust weather in North China. Therefore, it would be better to provide some references to explain why we should investigate the impacts of previous winter of NAO and ENSO on spring dust weather.

**Response:**

Thank you for the comments.

The standard deviation of the NAO peaks during December, January, February, and March. By analyzing the trend of the three-month average standard deviation, we observe that it is highest during the preceding winter. This indicates that the NAO exhibits stronger variability in boreal winter compared to other seasons (Figure R5 a). Similarly, ENSO also shows greater variation during boreal winter (Figure R5 b). Based on these findings, we have chosen to focus on the relationship between NAO, ENSO during the previous winter period, and spring dust activities over North China.

Previous studies have found that previous NAO and ENSO play important role in impacting the following climate over North China, particular the cross-seasonal impacts (e.g., Zheng et al., 2016; Feng et al., 2019; Sun et al., 2021). We have examined the role of previous autumn, winter and simultaneous spring NAO and ENSO on the spring dust aerosols over North China, and it is found the influences of NAO and ENSO on the spring dust aerosols are most significant in the previous winter (Figure R5 c-h). Thus, the role of previous winter NAO and ENSO on the spring dust aerosols over North China are discussed in the present work.

[Figure]

**Figure R5**. The monthly standard deviation of the (a) NAOI and (b) Niño3.4 index, respectively. Black line represents the trend of the three-month average standard deviation. Spatial distribution of correlation coefficients between the previous autumn NAOI and spring dust content (c). (d) As in (c), but with Niño3.4 index. (e-f) and (g-h), As in (c-d), but for the previous winter and simultaneous spring NAOI and Niño3.4 index. The black box represents North China. Stippled areas are statistically significant at the 0.1 level.

3. In the paper, the authors primarily discuss the effect of NAO and ENSO negative phases on the dust activities over North China. However, given the various phases combinations between these two factors, a more detailed explanation as to why only the negative-negative combinations are considered.

**Response:**

Thank you for the comments.

As shown in Fig. 2 of the manuscript, we have examined the role of different phases of NAO and ENSO on the dust aerosols over North China, and it is found the influences of the positive phases of NAO and ENSO on the dust aerosols are insignificant. "The results indicate that the relationship between NAO/ENSO and dust in North China also exhibits significant asymmetry, i.e., with weaker (stronger) correlations during positive (negative) phases of NAO and ENSO, where significant correlations only appear in the negative phases of NAO and ENSO" (Lines 216-219 in the revised manuscript).

The correlation coefficients between previous winter NAO, ENSO and spring dust aerosol content over North China under different phases are given in Table R1. It further explains that the influence of the negative phases of NAO and ENSO on the dust activities over North China is more significant than when they are in the positive phases. Based on the above discussion, we considered the effect of NAO and ENSO on dust aerosols over North China when they are in the negative phases.

[Figure]

**Figure 2 of Manuscript**. Spatial distribution of correlation coefficients between (a) positive and (c) negative NAOI values and dust content. (b) and (d), As in (a) and (c), respectively, but for the Niño3.4 index. Stippled areas are statistically significant at the 0.2 level.

**Table R1**. Correlation coefficients between the NAOI, ENSO index and regional average dust aerosol content over North China in spring under different phases. * indicates significant at the 0.1 level.

|  | Correlation coefficients |
|---|---|
| DJF_NAO+ & MAM _DUST | 0.05 |
| DJF_NAO- & MAM _DUST | -0.46[*] |
| DJF_ENSO+ & MAM _DUST | -0.16 |
| DJF_ ENSO + & MAM _DUST | -0.36[*] |

4. In Figures 1-2, the authors illustrate the relationship between NAO, ENSO and dust weather over North China through the spatial distribution of correlation coefficients, and that the relationship is only manifested when NAO and ENSO are in negative phases. A quantitative analysis is needed to further establish the robustness of the result.

**Response:**

Thank you for the comments.

We have adopted the reviewer's comment and added the quantitative analysis of NAO, ENSO on the dust aerosol concentration over North China. And we have added the description into the revised manuscript, as "Notably, North China is situated at the center of the maximum correlation, with correlation coefficients of -0.36 and -0.35 between NAO and ENSO, respectively" as shown in Lines 193-194. As well as, the correlation coefficients of NAO, ENSO during different phases and dust aerosol concentration over North China in Figure 2 are described as "Based on the scatter distribution of spring dust index (SDI) under different phases of NAO and ENSO, it is noted that the correlation coefficients between NAOI and SDI during the positive and negative phases of NAO are -0.46 and -0.05, respectively, indicating that the significant influence of NAO on the dust in North China mainly occurs during its negative phase. Similarly, the correlation distribution between the ENSO and SDI also shows that the

influence of ENSO is more pronounced during its negative phase" (Lines 222-227 in the revised manuscript).

5. From Fig 3 and Table 1, it is evident that there are two types when NAO and ENSO are in their negative phases: negative phases of the NAO and ENSO, and negative phases of the NAO and ENSO occurring separately (remove the years with concurrent negative phases of NAO and ENSO). Furthermore, the subsequent composite analyses in the study, focus on the cases with negative phases of the NAO and ENSO. The authors should explain why they have made this choice.

**Response:**

Thank you for the comments.

As shown in Figure 3c of manuscript, when the NAO is in its negative phase, including alone occurrence and in conjunction with negative phase of ENSO, the anomalous values of dust content over North China is 8.32 mg·m$^{-2}$ and 16.21 mg·m$^{-2}$, respectively. Similarly, the anomalous dust content over North China is 14.88 mg·m$^{-2}$ and 19.40 mg·m$^{-2}$ for the case of ENSO (Figure R6).

The above results show that no matter what kind of NAO and ENSO negative phase occurs, the increase in dust aerosol concentration over North China can be observed. The samples in the case of NAO negative phase is 8 and 15, respectively, and it is of 9 and 16 in the case of ENSO. In order to not only retain the characteristics of the negative phases of NAO and ENSO, but also make our results statistically characteristic, we selected the case of enough samples of the negative phases of NAO and ENSO to consider. We have added the description into the revised manuscript, as "To enhance the robustness of statistical analysis, we aim to select representative sample. Consequently, we focus on cases exhibiting negative phases of both the NAO and ENSO", as shown in Lines 244-246.

[Figure]

**Figure R6**. (a) Spring dust content over North China during the negative NAO, negative ENSO phases, and concurrent negative phases of NAO and ENSO. Transparent bars represent negative phases of the NAO and ENSO (unit: mg m$^{-2}$).

6. In Figs 4 c, f, and i, the variations in the near-surface wind field caused by anomalies of Siberian High, lead to dust emissions from the source areas. However, the depiction of the wind field anomalies appears unclear. It is recommended to modify the Figs to highlight the variations in the wind field.

**Response:**

Thank you for the comments and suggestions.

We have revised the figures to highlight the variations in the near-surface wind field caused by anomalies of Siberian High (SH). "In the sea level pressure field, during the negative phase of the NAO, the intensification of the SH typically accompanied with strong northerlies and dry conditions, favoring for the transport of dust, thereby supplying abundant material sources for dust activities in North China (Figure 4c of Manuscript). In the negative ENSO phase, more significant cyclonic circulation anomalies occur over the Western North Pacific (Figure 4f of Manuscript). When both the NAO and ENSO are in their negative phases, the strength and influence extent of the SH are more pronounced compared to that when the NAO sole is in negative phase (Figure 4i of Manuscript)" (Lines 302-312 in the revised manuscript).

[Figure]

**Figure 4 of Manuscript**. Upper, (a) 200 hPa zonal wind anomalies (shading, unit: m s-1), (b) 500 hPa geopotential height (shading, unit: gpm) and 850 hPa wind field anomalies (arrows, unit: m s-1), (c) sea-level pressure (shading, unit: Pa) and 1000 hPa wind field anomalies (arrows, unit: m s-1) during the negative NAO phases. Middle-Lower, as in the upper, but during the negative ENSO phases and concurrent negative phases of NAO and ENSO, respectively. Stippled areas and green arrows are statistically significant at the 0.2 level.

7. In Table 2, the value of correlation coefficients between the previous winter NATI and spring NATI are similar in scenarios of ENSO- phase (when the negative phase of ENSO occurs alone) and NAO- & ENSO- phase (when the negative phases of both NAO and ENSO co-occur). However, if there exists a synergistic effect of NAO and ENSO on the dust weather, the correlation in the scenario where both NAO and ENSO negative phases co-occur should be higher than when the negative phases of NAO and ENSO occur separately. The authors should have provided a more detailed explanation to clarify this point.

**Response:**

Thank you for the important comments.

It is seen that the correlation coefficients between the previous winter NATI and the subsequent spring NATI remain consistent across both scenarios (ENSO- phase,

NAO- & ENSO- phase), we would like to clarify this point into the following considerations.

The steps of NAO during the previous winter to affect the spring dust activities over North China divided into the following: 1) The NAO during the previous winter stimulates NAT; 2) The NAT can last from previous winter to spring due to the thermal persistence of the sea surface temperature; 3) The spring NAT modulates the circulation pattern over North China through teleconnection wave trains, which ultimately affects the spring dust activities over North China.

It is seen from Table 2 in the manuscript that although in the case of ENSO- phase and NAO- & ENSO- phase, the correlation coefficients of NATI in the previous winter and spring NATI are similar, both of which are 0.69. However, in the process of stimulating NAT by NAO in the previous winter, the correlations between NAO and NAT is higher during NAO- & ENSO- phase (0.66) than ENSO- phase (0.52). This suggests that the NAO significantly drives the NAT in the case of NAO- & ENSO- phase. The above discussion illustrates the synergistic effect of NAO and ENSO on the dust activities over North China (Lines 415-425 in the revised manuscript).

**Table 2 of Manuscript**. Correlation coefficients between the NAOI and NATI in three different categories. * indicates significant at the 0.1 level.

|  | DJF_NAO & DJF _NATI | DJF_NATI & MAM_NATI |
|---|---|---|
| NAO⁻ phase | $0.41^*$ | $0.51^*$ |
| ENSO⁻ phase | $0.52^*$ | $0.69^*$ |
| NAO⁻ & ENSO⁻ phase | $0.66^*$ | $0.69^*$ |

8. The main mechanism for the impact of the winter NAO on the spring dust is the maintenance of the North Atlantic SST anomalies from winter to spring, consistent with previous findings (Chen et al. 2020; Wu and Chen 2020; Song et al. 2022). Several discussions could be added.

**Response:**

Thank you for the comments.

We have quoted the work into the revised manuscript, as "which allows the previous NAO signal to exert a long-term influence on the subsequent weather and

climate in China (e.g., Chen et al., 2020; Wu and Chen, 2020; Song et al., 2022)", as shown in Lines 372-374.

9.  There are lots of clerical errors, i.e.,

Line 17-18, sea surface temperatures (SST) in the North Atlantic

Line 220, with regard to the description of the graphs, there may be some errors that"(b) and (d) As in (a) and (b) "-> "(c) and (d) As in (a) and (b)".

The authors should carefully check the whole manuscript.

**Response:**

Thanks to the reviewer for the comments. We have checked the whole manuscript and revised the errors.

**References:**

➤ Barnston, A. G., and Livezey, R. E.: Classification, seasonality and persistence of low-frequency atmospheric circulation patterns. Mon. Wea. Rev., 115, 1083–1126, https://doi.org/10.1175/1520-0493(1987)115<1083:CSAPOL>2.0.CO;2, 1987.

➤ Chen et al. 2020: Strengthened connection between springtime North Atlantic Oscillation and North Atlantic tripole SST pattern since the late-1980s. Journal of Climate, 35(5), 2007-2022.

➤ Feng, J., Li, J. P., Liao, H., and Zhu, J. L.: Simulated coordinated impacts of the previous autumn North Atlantic Oscillation (NAO) and winter El Niño on winter aerosol concentrations over eastern China, Atmos. Chem. Phys., 19, 10787–10800, https://doi.org/10.5194/acp-19-10787-2019, 2019.

➤ Hurrell, J. W.: Decadal Trends in the North Atlantic Oscillation: Regional Temperatures and Precipitation, Science, 269, 676–679, https://doi.org/10.1126/science.269.5224.676, 1995.

➤ Jones, P. D., Jonsson, T., and Wheeler, D.: Extension to the North Atlantic Oscillation using early instrumental pressure observations from Gibraltar and South-West Iceland. Int. J. Climatol., 17, 1433–1450, https://doi.org/10.1002/(SICI)1097-0088(19971115)17:13<1433::AID-JOC203>3.0.CO;2-P, 1997.

➤ Li, J. P., and Wang, J. X. L.: A new North Atlantic Oscillation index and its variability, Adv. Atmos. Sci., 20, 661–676, https://doi.org/10.1007/BF02915394, 2003.

➤ Moses, T., Kiladis, G. N., Diaz, H. F., and Barry, R. G.: Characteristics and frequency of reversals in mean sea level pressure in the North Atlantic sector and their relationship to long-term temperature trends. Int. J. Climatol., 7, 13–30, https://doi.org/10.1002/joc.3370070104, 1987.

➤ Najibi, N., Devineni, N., and Lall, U.: Compound Continental Risk of Multiple Extreme Floods in the United States. Geophys. Res. Lett., 50, e2023GL105297. https://doi.org/10.1029/2023GL105297, 2023.

➤ Parry, S., Lavers, D., Wilby, R., Prudhomme, C., Wood, P., Murphy, C., and OConnor, P.: Abrupt drought termination in the British–Irish Isles driven

by high atmospheric vapour transport, Environ. Res. Lett., 18, 104050, https://doi.org/10.1088/1748-9326/acf145, 2023.

➢ Rogers, J. C.: The association between the North Atlantic Oscillation and the Southern Oscillation in the Northern Hemisphere. Mon. Wea. Rev., 112, 1999-2015, https://doi.org/10.1175/1520-0493(1984)112<1999:TABTNA>2.0.CO;2, 1984.

➢ Song, L.-Y., et al, 2022: Distinct evolutions of haze pollution from winter to following spring over the North China Plain: Role of the North Atlantic sea surface temperature anomalies. Atmos. Chem. Phys., 22, 1669–1688.

➢ Sun, L. Y., Yang, X. Q., Tao, L. F., Fang, J. B., and Sun, X. G.: Changing Impact of ENSO Events on the Following Summer Rainfall in Eastern China since the 1950s, J. Climate, 34, 8105–8123, https://doi.org/10.1175/JCLI-D-21-0018.1, 2021.

➢ Wu and Chen, 2020: What leads to persisting surface air temperature anomalies from winter to following spring over the mid-high latitude Eurasia?. Journal of Climate, 33, 5861-5883.

➢ Wang, L., and Ting, M. F.: Stratosphere-Troposphere Coupling Leading to Extended Seasonal Predictability of Summer North Atlantic Oscillation and Boreal Climate, Geophys. Res. Lett., 49, e2021GL096362. https://doi.org/10.1029/2021GL096362, 2022.

➢ Zheng, F., Li, J. P., Li, Y. J., Zhao, S., and Deng, D. F.: Influence of the Summer NAO on the Spring-NAO-Based Predictability of the East Asian Summer Monsoon, J. Appl. Meteorol. Clim., 55, 1459–1476, https://doi.org/10.1175/JAMC-D-15-0199.1, 2016.

---

## Author Comment (AC2)

**Response to Comments of Reviewer 2**

**Manuscript number**: egusphere-2024-955

**Author(s)**: Falei Xu, Shuang Wang, Yan Li, and Juan Feng

**Title**: Synergistic effects of previous winter NAO and ENSO on the spring dust activities in North China

**General comments:**

Using multi-reanalysis datasets, the authors investigated the effects of the previous winter NAO and ENSO on the spring dust aerosols over North China. The pronounced influence of NAO and ENSO on dust aerosols was predominantly observed during their negative phases. Furthermore, this analysis examined meteorological conditions, atmospheric dynamics, and wave energy transport, elucidating the synergistic impacts of these negative phases on subsequent dust activities. The findings enhance our understanding of the formation mechanisms of dust events in North China. I recommend that this manuscript be accepted after minor revisions, as this study fits well within the scope of Atmospheric Chemistry and Physics.

**Response:**

Thanks to the reviewer for the helpful comments and suggestions. We have revised the manuscript seriously and carefully according to the reviewer's comments and suggestions. The point-to-point responses to the comments are listed as follows.

**Specific comments are as follows:**

1. The study primarily utilizes MERRA-2 reanalysis data for analyzing dust activities over North China. It is essential to assess whether the MERRA-2 data accurately captures dust activities in North China. Please provide further details on the reliability of the reanalysis.

**Response:**

Thank you for the comments.

Previous studies have demonstrated the accuracy and applicability of MERRA-2 reanalysis data for studying the evolution of dust events in Asia. "Its analytical results are similar to those obtained from MODIS, OMPS, CALIPSO, and Himawari-8 data (Kang et al., 2016; Yao et al., 2020; Wang et al., 2021)" (Lines 131-135 in the revised manuscript).

Additionally, we further employ the datasets from the China National Meteorological Centre, which include observations of floating dust, blowing dust, and dust storms, to validate the MERRA-2 reanalysis data. The frequency of dusty weather recorded at these stations has been converted into a Dust Index (DI) (Wang et al., 2008; Equations 1), effectively representing the concentration of dust aerosols.

$$DI = 9 \times DS + 3 \times BD + 1 \times FD \qquad (1)$$

Where DS, BD, and FD represent the frequency of dust storms, blowing dust, and floating dust, respectively. Additionally, DI denotes the concentration of dust aerosols at each station.

We found that the variations of the DI and MERRA-2 dust aerosols concentration during the four seasons all show similar spatial characteristics. Especially for the dust source in Northwest China and the spring dust aerosols over North China, the spatial distribution characteristics are relatively consistent (Figure R1). The above results indicate that the MERRA-2 aerosol reanalysis data can simulate the spatiotemporal distribution characteristics of dust aerosol concentration in China, which is applicable and effective for us to understand the variations in dust aerosol concentration in China.

[Figure]

**Figure R1**. (a-d) Seasonal distribution of DI from station data, (e-h) As in (a-d), but for dust column mass density from MERRA-2 reanalysis data during 1980-2018 (units: mg m$^{-2}$).

2. The preceding role of NAO and ENSO on the spring dusty weather over North China is investigated, and it is of interest why the preceding role is focused. And whether their simultaneous role in the dust content is significant or not.

**Response:**

Thank you for the comments.

Previous studies have found that previous NAO and ENSO play important role in impacting the following climate over North China, particular the cross-seasonal impacts (e.g., Zheng et al., 2016; Feng et al., 2019; Sun et al., 2021). Moreover, the one season ahead signals can provide as the useful predictors for the spring dust activities in North China.

The standard deviation of the NAO peaks during December, January, February, and March. By analyzing the trend of the three-month average standard deviation, we observe that it is highest during the preceding winter. This indicates that the NAO exhibits stronger variability in the winter compared to other seasons (Figure R2 a). Similarly, ENSO also shows greater variation during boreal winter (Figure R2 b). Based on these findings, we have chosen to focus on the relationship between NAO, ENSO during the pre-previous winter period, and spring dust activities over North China.

We have examined the role of previous autumn, winter and simultaneous spring NAO and ENSO on the spring dust aerosols over North China, and it is found the influences of NAO and ENSO on the spring dust aerosols are most significant in the previous winter (Figure R2 c-h). Thus, the role of previous winter NAO and ENSO on the spring dust aerosols over North China are discussed in the present work.

[Figure]

**Figure R2**. The monthly standard deviation of the (a) NAOI and (b) Niño3.4 index, respectively. Black line represents the trend of the three-month average standard deviation. Spatial distribution of correlation coefficients between the previous autumn NAOI and spring dust content (c). (d) As in (c), but with Niño3.4 index. (e-f) and (g-h) As in (c-d), but for the previous winter and simultaneous spring NAOI and Niño3.4 index. The black box represents North China. Stippled areas are statistically significant at the 0.1 level.

3. Dust aerosols are important components of atmospheric aerosols, alongside other constituents such as sulfates, nitrogen oxides, black carbon, and so on. Why did the authors choose the dust aerosols in North China as the research objects to be discussed and studied? Please explain and justify this point.

**Response:**

Thank you for the comments.

The dust aerosols are characterized by their significant role within the broader category of atmospheric aerosols. Dust aerosols originate from the mechanical breakdown of rocks and soil into fine particles, which are subsequently transported by the wind (Wang et al., 2018). They are noteworthy for their ability to influence climate systems by affecting solar radiation and cloud formation (e.g., Sokolik and Toon, 1996; Sassen et al., 2003; Zhang et al., 2019). Dust aerosols can pose a formidable threat to socio-economic development, natural ecological environment, as well as human health and safety (e.g., Zhao et al., 2020; Yin et al., 2021; Li et al., 2023). The study of dust aerosols is essential, because understanding their properties and dynamics helps us to better predict weather patterns, assess climate change impacts, and implement effective environmental and public health policies.

4. Line 36-37, "The Gobi Desert in East Asia, especially for the Mongolian Plateau and North China, is a major source of dust". Whether the author is trying to express the meaning of Northern China here, Northern China and North China are two different meanings, please confirm and revise.

**Response:**

Thank you for the comments. We have checked the whole manuscript and revised similar errors.

5. Line 77-78, the authors mentioned that "NAO and ENSO often co-occur and have complex interactions". As well as by citing previous work, the facts of a possible relationship between the two factors are enumerated. However, the authors have not thoroughly explored their relationship. It is suggested that further details be provided to enhance the understanding.

**Response:**

Thank you for the comments.

The relationship between the NAO and the ENSO remains unclear. Statistical analyses largely indicate no significant linear association between them. For instance, a correlation analysis of the NAO and ENSO indices during period 1950-2000 shows that their correlation coefficient is only 0.09, suggesting a weak linear correlation (Wang, 2002). Additionally, a significant correlation exists between the La Niña events in autumn and the positive phase of the NAO. However, this is not the case during El Niño events (Pozo-Vazquez et al., 2005).

Recent researchers have further detected the relationship between the NAO and the ENSO, particularly following the identification of two distinct types of ENSO events: the Eastern Pacific (EP) and Central Pacific (CP) El Niño events. It is suggested that the EP El Niño can transmit the Pacific signal to the North Atlantic through a subtropical bridge mechanism, potentially triggering a negative phase in the NAO. However, this relationship is not notably significant (Graf and Zanchettin, 2012). Moreover, the atmospheric circulation in the North Atlantic region reacts differently to the two types of La Niña events. During an EP La Niña, when the North Atlantic jet is weakened, the NAO tends to be in a negative phase. Conversely, CP La Niña would strengthen the North Atlantic jet, and the NAO is more likely to exhibit a positive phase (Zhang et al., 2015).

The above discussion suggests that there may be a nonlinear link between NAO and ENSO, and the relationship between them is still inconclusive and requires further study. Therefore, this paper analyzes the synergistic effect of NAO and ENSO on dust activities over North China.

6. In Figure 3 (a), it is notable that there is a point during the negative phase of the NAO that deviates from the majority of the points, potentially qualifying it as an outlier. If this point is removed from the sequence, it is important to verify whether the relationship between the NAO and dust aerosol content remains robust.

**Response:**

Thank you for the important comments.

It is important to consider whether the influence of the negative phase of the NAO on dust activities in North China persists after removing the outlier. When the outlier is excluded, we observe a reduction in the correlation between the two factors, yet a significant correlation remains and passes the 0.2 statistical significance test (Figure R3). This suggests that the impact of the NAO on dust activities in North China is robust during its negative phase.

[Figure]

**Figure R3**. Scatterplots of the spring dust content in North China against previous winter (a) NAOI and (b) NAOI (Remove Outlier). Also shown are lines of best fit for positive and negative NAO/Niño3.4 index values and correlation coefficients (R), slope (slope), * indicates significant at the 0.2 level.

7. In Figure 5, by describing the precipitation and humidity fields under different scenarios, the authors illustrate the synergistic effect of NAO and ENSO on dust activities in North China. However, the large values of the variables in the graphs do not seem to be well highlighted. Consider modifying the color-coded intervals to enhance the reader's understanding of the section.

**Response:**

Thank you for the comments and suggestions. We have revised the figures to highlight large values of the precipitation and humidity fields under different scenarios. "When the NAO is in its negative phase, humidity in the spring dust source regions and North China generally reduced, particularly in areas near the dust source regions, indicating that these areas are conducive to dust transport and prone to causing dust weather in North China (Figure 5a of Manuscript). As for the precipitation, there is more spring precipitation in the northwest region of China, while precipitation in the Mongolia and the North China is relatively less (Figure 5b of Manuscript). In the negative ENSO phase, the variation in humidity is similar to

that during the negative NAO phase, but with a greater amplitude (Figure 5c of Manuscript), indicating that ENSO has a stronger impact on the humidity conditions in North China. Moreover, the precipitation shows a significant abnormal decrease over Mongolia and North China, which is highly conducive to dust activities and the generation of dust weather (Figure 5d of Manuscript). When both the NAO and ENSO are in the negative phases, the humidity anomalies in the dust source regions and North China are more intense than the individual factor (Figure 5e of Manuscript). The variation in precipitation are similar to those in humidity, the reduction in precipitation in the dust source regions and North China exceeds the sole role (Figure 5f of Manuscript)" (Lines 335-348 in the revised manuscript).

[Figure]

**Figure 5 of Manuscript**. Upper, composite percentage anomalies of (a) specific humidity and (b) precipitation during negative NAO phases. Middle-Lower, as in the upper, but during negative ENSO phases and concurrent negative phases of NAO and ENSO, respectively. Stippled areas are statistically significant at the 0.2 level.

**Reference:**

➢ Feng, J., Li, J. P., Liao, H., and Zhu, J. L.: Simulated coordinated impacts of the previous autumn North Atlantic Oscillation (NAO) and winter El Niño on winter aerosol concentrations over eastern China, Atmos. Chem. Phys., 19, 10787–10800, https://doi.org/10.5194/acp-19-10787-2019, 2019.

➢ Graf, H.-F. and Zanchettin, D.: Central Pacific El Niño, the "subtropical bridge," and Eurasian climate, J. Geophys. Res.-Atmos., 117, https://doi.org/10.1029/2011JD016493, 2012.

➢ Kang, L. T., Huang, J. P., Chen, S. Y., and Wang, X.: Long-term trends of dust events over Tibetan Plateau during 1961-2010, Atmos. Environ., 125, 188-198, https://doi.org/10.1016/j.atmosenv.2015.10.085, 2016.

➢ Li, Y., Xu, F. L, Feng, J., Du, M. Y., Song, W. J., Li, C., and Zhao, W. J.: Influence of the previous North Atlantic Oscillation (NAO) on the spring dust aerosols over North China, Atmos. Chem. Phys., 23, 6021–6042, https://doi.org/10.5194/acp-23-6021-2023, 2023.

➢ Pozo-Vazquez, D., Gamiz-Fortis, S. R., Tovar-Pescador, J., Esteban-Parra, M. J., and Castro-Diez, Y.: North Atlantic winter SLP anomalies based on the autumn ENSO state, J. Clim., 18, 97–103, 2005.

➢ Sassen, K., DeMott, P. J., Prospero, J. M., and Poellot, M. R.: Saharan dust storms and indirect aerosol effects on clouds: CRYSTAL-FACE results, Geophys. Res. Lett., 30, 1633, https://doi.org/10.1029/2003GL017371, 2003.

➢ Sokolik, I. N. and Toon, O. B.: Direct radiative forcing by anthropogenic airborne mineral aerosols, Nature, 381, 681–683, https://doi.org/10.1038/381681a0, 1996.

➢ Sun, L. Y., Yang, X. Q., Tao, L. F., Fang, J. B., and Sun, X. G.: Changing Impact of ENSO Events on the Following Summer Rainfall in Eastern China since the 1950s, J. Climate, 34, 8105–8123, https://doi.org/10.1175/JCLI-D-21-0018.1, 2021.

➢ Wang C. Z.: Atlantic Climate Variability and Its Associated Atmospheric Circulation Cells, J. Climate, 15, 1516-1536, https://doi.org/10.1175/1520-0442(2002)015<1516:ACVAIA>2.0.CO;2, 2002.

➢ Wang, T. H., Tang, J. Y., Sun, M. X., Liu, X. W., Huang, Y. X., Huang, J. P., Han, Y., Cheng, Y. F., Huang, Z. W., and Li, J. M.: Identifying a transport mechanism of dust aerosols over South Asia to the Tibetan Plateau: A case study, Sci. Total Environ., 758, 11, https://doi.org/10.1016/j.scitotenv.2020.143714, 2021.

➢ Wang, X., Huang, J. P., Ji, M. X., and Higuchi, K.: Variability of East Asia dust events and their long-term trend, Atmos. Environ., 42, 3156-3165, https://doi.org/10.1016/j.atmosenv.2007.07.046, 2008.

➢ Wang, X., Liu, J., Che, H. Z., Ji, F., and Liu, J. J.: Spatial and temporal evolution of natural and anthropogenic dust events over northern China, Sci. Rep., 8, 2141, https://doi.org/10.1038/s41598018-20382-5, 2018.

➢ Yao, W. R., Che, H. Z., Gui, K., Wang, Y. Q., and Zhang, X. Y.: Can MERRA-2 Reanalysis Data Reproduce the Three-Dimensional Evolution Characteristics of a Typical Dust Process in East Asia? A Case Study of the Dust Event in May 2017, Remote Sens., 12, 18, https://doi.org/10.3390/rs12060902, 2020.

➢ Yin, Z. C., Wan, Y., Zhang, Y. J., and Wang, H. J.: Why super sandstorm 2021 in North China? Natl. Sci. Rev., 9, nwab165, https://doi.org/10.1093/nsr/nwab165, 2021.

➢ Zhang, C. X., Liu, C., Hu, Q. H., Cai, Z. N., Su, W. J., Xia, C. Z., Zhu, Y. Z., Wang, S. W., and Liu, J. G.: Satellite UV-Vis spectroscopy: implications for air quality trends and their driving forces in China during 2005–2017, Light-Sci. Appl., 8, 100, https://doi.org/10.1038/s41377-019-0210-6, 2019.

➢ Zhang, W. J., Wang, L., Xiang, B. Q., Qi, L., and He, J. H.: Impacts of two types of La Nina on the NAO during boreal winter, Clim. Dyn., 44, 1351-1366, https://doi.org/10.1007/s00382-014-2155-z, 2015.

➢ Zhao, C. F., Yang, Y. K., Fan, H., Huang, J. P., Fu, Y. F., Zhang, X. Y., Kang, S. C., Cong, Z. Y., Letu, H., and Menenti, M.: Aerosol characteristics and impacts on weather and climate over the Tibetan Plateau, Natl. Sci. Rev., 7, 492–495, https://doi.org/10.1093/nsr/nwz184, 2020.

➢ Zheng, F., Li, J. P., Li, Y. J., Zhao, S., and Deng, D. F.: Influence of the Summer NAO on the Spring-NAO-Based Predictability of the East Asian Summer

Monsoon, J. Appl. Meteorol. Clim., 55, 1459–1476, https://doi.org/10.1175/JAMC-D-15-0199.1, 2016.

---

## Author Response (AR2)

**Response to Comments of Editor**

**Manuscript number**: egusphere-2024-955

**Author(s)**: Falei Xu, Shuang Wang, Yan Li, and Juan Feng

**Title**: Synergistic effects of previous winter NAO and ENSO on the spring dust activities in North China

**General comments:**

Please clearly highlight which paragraphs are new or modified in the revised version. Moreover, line numbers should be referred more accurately.

**Response:**

We have seriously and carefully revised the manuscript, highlighted the modified paragraphs and labeled the line numbers in the revised version. The point-to-point responses to the comments are listed as follows.

**Specific comments to Response to reviewers:**

**Response to R1:**

1. The paragraph on NAOIs comparison reported here does not match with the revised version of the manuscript. Please: 1) modify the revised manuscript accordingly; 2) provide a reference to the source of data you use for the comparison.

**Response:**

Thanks for the suggestions.

We have revised the paragraph on NAOIs comparison in the revised manuscript (Lines 161-164). The relevant reference has been included, and the source of data for the NAOI is included in the "Code and data availability" (Lines 598-602).

2. The analysis shown in Fig. R5 could be added to the main text, to justify the choice of analysing DJF vs MAM.

**Response:**

Thanks for the suggestions.

We have adopted the editor's suggestion and include the figure (Figure 2 in the revised manuscript) and corresponding descriptions into the revised manuscript (Lines 202-212).

3. The paragraph reported here does not match with the revised version of the text, please modify accordingly. Also, the figure is not very much improved, please try to reduce the number of plotted vectors and increase the size.

**Response:**

Thanks for the suggestions.

We have revised the paragraph in the revised manuscript (Lines 336-348). And we have revised the figure to highlight the variations in the near-surface wind field (Figure 6 in the revised manuscript, also seeing in Figure R1).

[Figure]

**Figure R1.** Upper, the composite anomalies of (a) 200 hPa zonal wind (shading, unit: m·s⁻¹), (b) 500 hPa geopotential height (shading, unit: gpm) and 850 hPa wind field (arrows, unit: m·s⁻¹), (c) sea-level pressure (shading, unit: Pa) and 1000 hPa wind field (arrows, unit: m·s⁻¹) during the negative NAO phases. Middle-Lower, as in the upper, but during the negative ENSO phases and co-occurred negative phases of NAO and ENSO, respectively. The green box represents North

China. Only wind anomalies statistically significant at the 0.1 level are shown. Thick and fine stippled areas are statistically significant at the 0.05 and 0.1 level, respectively.

**Response to R2:**

1. The comparison between MERRA and the Dust Index might be included in the manuscript.

**Response:**

Thanks for the suggestions.

We have included the relevant figure (Figure 1 in the revised manuscript, also seeing in Figure R2) as well as the corresponding descriptions in the revised manuscript (Lines 126-140).

[Figure]

**Figure R2**. (a-d) Spatial distribution of seasonal mean DI based on station data, (e-h) As in (a-d), but for dust column mass density based on MERRA-2 (units: mg·m$^{-2}$). The green box in (a) and (e) represents North China. The green lines represent the Yellow River (northern one) and the Yangtze River (southern one), respectively.

2. It is not clear the difference in the notations North and northern China: please elaborate more the response, and modify the manuscript accordingly.

**Response:**

Thanks for the comments and suggestions.

The North China, as a crucial center of politics, economy, and population, refers to the extent of 30°-40°N, 105°-120°E. The northern China (north of 30°N) refers to a broader extent, which encompasses North China, northeast China, Hexi Corridor, and northwest China.

**General comments to the manuscript:**

1. Significance level should be homogeneous through the text, and set to 0.05. You could set an additional threshold at 0.10, defining weak significance. P-value at 0.2 is too low, and cannot be used to define statistical significance. Furthermore, computing correlations for 7-year time series (negative NAO & ENSO) is questionable. I recommend to revise the whole correlation analysis and adjust your discussion and conclusions accordingly.

**Response:**

Thanks for the important comments.

We have modified the significance level in the revised version. Thick and fine stippled areas are statistically significant at the 0.05 and 0.1 level, respectively.

The MERRA-2 data starts from 1980, thus the period 1980-2022 is selected in this study. The 0.5 standard deviation threshold is used to select the anomalous events of the NAO and ENSO. There are 15 (16) years of negative NAO (ENSO), and there are 7 co-occurrence years of negative NAO and ENSO. Then the composite analysis is used to detect the associated circulation anomalies along with three situations, i.e., negative NAO, negative ENSO, and both the NAO and ENSO are negative.

The co-occurrence of negative NAO and ENSO takes up to 17% of the whole study period. We realize the samples are not long enough, however, it is subjected to the longevity of the datasets. We have included the above discussions into the revised

manuscript (Lines 582-587), and it is worthy to further examine their joint impacts by models.

2. The use of correlation and composite analysis is not very clear to me. On the one hand, when computing correlations during negative phases of NAO/ENSO, you are assessing the impact of negative phases of different magnitude on the dust content/atmospheric variables/SST, i.e., you are studying the variability of the relationship. On the other hand, when computing composites, you are assessing the overall impact of the negative phases, i.e., you are studying the impact on the mean values. The specific objectives of these two complementary approaches should be clarified.

**Response:**

Thanks for the important comments. We would like to clarify this in the following points:

1) The correlation and composite analysis serve different purposes, but both of them aim to understand the impact of the NAO and ENSO on the dust content over North China.

2) Correlation analysis is used to explore the relationship between NAO/ENSO and dust content over North China. It is found that both NAO and ENSO exert significant effects on dust activities in North China, especially during their negative phases. These results indicate that the impacts of previous winter NAO and ENSO on the spring dust content in North China exhibit asymmetrical characteristics, significant effects mainly manifested during their negative phases.

3) Composite analysis is widely used to explore the synergistic effects of climatic variabilities on the regional climate anomalies (e.g., Wang et al., 2023; Dong et al., 2024; Tang et al., 2024). It is found that when both NAO and ENSO are in negative phases (7 co-occurrence years minus climatology) simultaneously, their impacts on dust activities in North China is greater than that of either NAO (15 NAO negative years minus climatology) or ENSO (16 ENSO

negative years minus climatology) is in the negative phase. This indicates that NAO and ENSO in negative phases have a synergistic effect on dust activities in North China.

We have added the correlation analysis, composite analysis, and significance testing used into the "Methods" section of revised manuscript (Lines 170-174).

3. English language needs a deep revision.

Thanks for the comments. We have revised the manuscript to enhance the clarity and conciseness of the English.

**Specific comments to the manuscript:**

1. Line 10: "by impacting".

**Response:**

Yes, done.

2. Line 11: "Based on the atmospheric and oceanic datasets during 1980-2022…" please rephrase.

**Response:**

We have revised it, as shown "Based on the reanalysis datasets during 1979-2022…" in Lines 11.

3. Line 14: "exert significant effects in influencing…" please rephrase.

**Response:**

We have revised it, as shown "exert significant effects on dust activities in North China…" in Lines 14.

4. Lines 13-17: These two sentences redundant, please rephrase.

**Response:**

We have revised it, as shown "It is found that both the NAO and ENSO exert significant effects on dust activities in North China, especially during their negative

phases. When both of them are in the negative phases, their combined impacts on the dust activities exceeding that of either factor individually" in Lines 13-16.

5. Line 18: "associated".

**Response:**

Yes, done.

6. Line 18: "Owing to the persistence of SST…" please rephrase.

**Response:**

We have revised it, as shown "These SST anomalies can persist to the following spring due to their inherent persistence" in Lines 18-19.

7. Line 24: What is "dust weather"? Here and across the text.

**Response:**

The "dust weather" has been replaced into "dust activities" throughout the manuscript.

8. Line 39: What is "dust disasters"? Here and across the text.

**Response:**

We have revised this and "dust activities" is used in the revised manuscript.

9. Lines 110-117: There are many redundancies in this paragraph, please rephrase to streamline it.

**Response:**

We have revised it, as shown "The synergistic effects of NAO and ENSO significantly influence the climate in China, but their synergistic effects on the dust activities over North China and the mechanisms involved remain unclear. This study will investigate these effects on dust activities over North China, providing a scientific foundation for predicting dust activities in China" in Lines 110-113.

**10. Line 141: Please define how the last winter of your time series is defined.**

**Response:**

The winter is defined as December-February (December-January-February, DJF), with the winter 1979 corresponding to the average of December in 1979, January and February in 1980 (Lines 147-149). The winter NAO and ENSO indices are during 1979-2021, and the spring dust are during 1980-2022, to highlight the preceding impacts of previous winter on the following spring (Lines 168-170).

**11. Lines 143-144: By removing the linear trend, you are not "enhancing the investigation of the NAO-ENSO-dust relationship", you are actually focusing your analysis on the interannual variability. Please rephrase.**

**Response:**

We have revised it, as shown "To focus the investigation into the interannual variability, the linear trends of all variables were removed" in Lines 149-150.

**12. Line 155: Why is ENSO defined in the period 1980-2022?**

**Response:**

This is a clerical error, the winter Niño 3.4 index is from 1979 winter to 2021 winter.

**13. Line 192: How these correlation coefficients are computed?**

**Response:**

The correlation coefficients are computed between the areal mean spring dust content over North China and previous winter NAOI and Niño 3.4 index. We have made the corresponding revisions in the text, as shown in Lines 229-230.

**14. Figure 1: What do the green lines represent? Same comment for Figures 2 and 5.**

**Response:**

The green lines in (c-h) represent the Yellow River (northern one) and the Yangtze River (southern one), respectively.

15. Figure 3: Panel (c) is rather puzzling: from Table 1 we see that there are three possible situations, that is negative NAO (15 years), negative ENSO (16 years), negative NAO and ENSO (7 years). So, I don't understand what the white bars represent (and why they are different from the green one). Please clarify in the caption and in the text. Can you also add uncertainty bars to the plot?

**Response:**

In order to access the synergetic impacts of negative NAO/ENSO phases on the spring dust activities over North China, three scenarios are considered, i.e., solo negative NAO (red in Figure R3), solo negative ENSO (blue in Figure R3), and both the NAO and ENSO are in negative phases (green in Figure R3). During period 1979-2022, there are 15 years in which the NAO is in the negative phase, and 7 of these 15 years are when both the NAO and ENSO are simultaneously in negative phases. The white bar (labeled as NAO) is the dust content anomalies during the 15 negative NAO years, and red bar is the anomalies for the solo NAO negative years (based on 8 years). The white bar (labeled as ENSO) is the dust content anomalies during the 16 negative ENSO years, and blue bar is the anomalies for the solo ENSO negative years (based on 9 years). The green bar (labeled as NAO&ENSO) is the dust content anomalies when both the NAO and ENSO are in negative phases (i.e., 7 co-occurrence years) (Figure 5 in revised manuscript, also seeing in Figure R4).

We have adopted the editor's comments by adding the uncertainty bars to the plot and rewriting the figure captions.

[Figure]

**Figure R3**. The different scenarios of negative phases of NAO and ENSO.

[Figure]

**Figure R4**. Scatterplots of the spring dust content in North China against previous winter (a) NAOI and (b) Niño3.4 index. Also shown are lines of best fit for positive and negative NAO/Niño3.4 index values and correlation coefficients (R), slope (slope), * indicates significant at the 0.1 level. (c) Spring dust content over North China during the negative NAO, negative ENSO phases, and concurrent negative phases of NAO and ENSO (unit: mg·m⁻²). White bars represent negative phases of the NAO and ENSO, red and blue bars indicate solo negative NAO and ENSO years, and green bar is the negative NAO and ENSO co-occurring years.

16. Line 262: How do you estimate this value? Is it just an approximate value you derive from the map? Do you compute a spatial average? Same comments for Lines 264 and 270.

**Response:**

This value is an approximate value from the map. Since the significant areas are different in the three scenarios (i.e., negative NAO, negative ENSO, and both the NAO and ENSO are in negative phases), the areal mean anomalous values in these three scenarios are incomparable. We have deleted the quantitative description of the anomalous circulation field in the revised manuscript.

17. Line 270: In comparing the zonal wind anomalies, do you assess statistical significance? Specifically, +3 m/s anomaly is significantly stronger than +1.5 and +2.5 m/s?

**Response:**

We have used two-tailed Student's *t*-test to assess the statistical significance of the composite anomalies.

The significant areas are different under different situations. We have deleted the quantitative values in the revised manuscript.

18. Line 274: Same comments for this paragraph: 1) How geopotential anomalies are estimated? 2) Significance in the differences has been estimated?

**Response:**

The two-tailed Student's *t*-test is used to estimate the statistical significance for the composite analysis. We have deleted the quantitative values in the revised manuscript.

19. Figures 4 and 5: Please add a box to highlight your study region.

**Response:**

Yes, done.

20. Line 370: Why Figure 7 is mentioned before Figure 6?

**Response:**

This is a clerical error, we have revised it.

21. Line 374: What is the rationale of the NATI definition? The SST patterns in Figure 6 resembles very much to a positive Atlantic Multidecadal Oscillation phase.

**Response:**

The correlation between spring dust content in North China and SST exhibits a "+ - +" tripole pattern in the North Atlantic (Figure 8 in the revised manuscript, also seeing in Figure R5), similar to the tripole SST anomalies induced by NAO (Visbeck et al., 2001; Wu et al., 2009). Therefore, the North Atlantic tripole index (NATI) is defined to

depict the characteristics of SST anomalies (Equations 4-7 in the revised manuscript, also seeing in Equations 1-4).

[Figure]

**Figure R5.** (a) Spatial distribution of the correlation coefficients between the spring SDI and simultaneous SST. (b)-(c) As in (a), but for the positive and negative phase of SDI. Thick and fine stippled areas are statistically significant at the 0.05 and 0.1 level, respectively. The black box represents NATI.

$$SST_A = [15-25°N, 32-20°W] \tag{1}$$
$$SST_B = [22-32°N, 75-60°W] \tag{2}$$
$$SST_C = [50-60°N, 50-32°W] \tag{3}$$
$$NATI = SST_B - \frac{1}{2}(SST_A + SST_C) \tag{4}$$

The NATI corresponds to positive SST anomalies over high and low latitude areas of the North Atlantic, with negative SST anomalies over middle latitude areas of the North Atlantic (Figure R5 a). The sign of SST anomalies in the high and low latitudes is opposite to that in the mid-latitudes. In contrast, the positive AMO is associated with positive SST anomalies over most of the North Atlantic, with stronger anomalies in the subpolar region and weaker anomalies in the tropics (Figure R6 a).

The NATI is insignificant correlated with interdecadal variation of AMOI with a coefficient of -0.24. Moreover, we further calculated the spatial correlation between the SST anomalies associated with the NATI and AMOI within North Atlantic (10°-70°N, 75°-10°W), with a coefficient of -0.25 and -0.15 for the positive and negative NATI

phases, respectively. That is both the NAT and AMO shows insignificant relationship in both temporal and spatial distributions. Additionally, the relationship between SDI and NATI (with a significant correlation coefficient of -0.33) is stronger than that between SDI and AMOI (correlation coefficient of 0.06) (Figure R7). This point indicates the AMO shows insignificant relationship with the SDI.

This study did not discuss the potential impacts of interdecadal signals, such as the AMO, on dust activities in China. The interdecadal variations of dust activities over China as well as its connection to the interdecadal climatic variabilities will be discussed in future work. We have included the above discussions into the revised manuscript (Lines 587-590).

[Figure]

**Figure R6**. (a) Observed AMO SST pattern, derived by regressing detrended North Atlantic annual mean SST anomalies on the observed AMO index for the period 1870-2015 (From the Climate Data Guide, NCAR).

[Figure]

**Figure R7**. (a) Normalized time series of spring SDI (black line) and NATI (red line) during 1980-2022. (b) As in (a), but for the AMOI (blue line). * indicates significant at the 0.05 level.

22. Figures 8, 9 and 10: Please explain why regression fields are multiplied by -1.

**Response:**

The impact of NAO and ENSO on the dust activities over North China is manifested in their negative phases, both of them are negatively correlated with the dust content. Therefore, to give a direct comparison between the NAO&ENSO associated circulation anomalies and the climatology, the regression fields are multiplied by -1. This method is widely used in the climate analysis to show a more direct comparison when two variables are negatively correlated (e.g., Larkin et al.,2002; Gong and Luo, 2017; Yao et al., 2017). For example, Yao et al. (2017) demonstrated the regression fields of winter mean 500hPa height and surface air temperature anomalies relative to Arctic sea ice anomalies during the period 1979-2011, and multiplied by -1 to facilitate a direct comparison.

**Reference**

Dong, W., Jia, X. J., Li, X. M., and Wu, R. G.: Synergistic effects of Arctic amplification and Tibetan Plateau amplification on the Yangtze River Basin heatwaves, npj Climate and Atmospheric Science, 7, 150, https://doi.org/10.1038/s41612-024-00703-4, 2024.

Gong, T. T and Luo, D. H.: Ural Blocking as an Amplifier of the Arctic Sea Ice Decline in Winter. J. Climate, 30, 2639-2654. https://doi.org/10.1175/JCLI-D-16-0548.1, 2017.

Larkin, N. K and Harrison, D. E.: ENSO Warm (El Niño) and Cold (La Niña) Event Life Cycles: Ocean Surface Anomaly Patterns, Their Symmetries, Asymmetries, and Implications, J. Climate, 15, 1118-1140. https://doi.org/10.1175/1520-0442(2002)015<1118:EWENOA>2.0.CO;2, 2002.

Tang, X. X and Li, J. P.: Synergistic effect of boreal autumn SST over the tropical and South Pacific and winter NAO on winter precipitation in the southern Europe. npj Climate and Atmospheric Science, 7, 78, https://doi.org/10.1038/s41612-024-00628-y, 2024.

Visbeck, M. H., Hurrell, J. W., Polvani, L., and Cullen, H. M. The North Atlantic Oscillation: Past, present, and future. Proc. Natl. Acad. Sci. USA., 98, 12876-12 877, https://doi.org/10.1073/pnas.231391598, 2001.

Wang, H., Li, J. P., Zheng, F., and Li, F.: The synergistic effect of the summer NAO and northwest pacific SST on extreme heat events in the central–eastern China, Clim. Dyn., 61, 4283–4300, https://doi.org/10.1007/s00382-023-06807-6, 2023.

Wu, Z. W., Wang, B., Li, J. P., and Jin, F. F.: An empirical seasonal prediction model of the east Asian summer monsoon using ENSO and NAO, J. Geophys. Res.-Atmos., 114, 2009JD011733, https://doi.org/10.1029/2009JD011733, 2009.

Yao, Y., Luo, D. H., Dai, A. G., and Simmonds L.: Increased Quasi Stationarity and Persistence of Winter Ural Blocking and Its Impact on East Asian Climate, J. Climate, 30, 3549–3568. https://doi.org/10.1175/JCLI-D-16-0261.1, 2017.

**Response to Comments of Reviewer 1**

**Manuscript number**: egusphere-2024-955

**Author(s)**: Falei Xu, Shuang Wang, Yan Li, and Juan Feng

**Title**: Synergistic effects of previous winter NAO and ENSO on the spring dust activities in North China

**General comments:**

This study investigated the impacts of preceding boreal winter North Atlantic Oscillation (NAO) and El Niño-Southern Oscillation (ENSO) on the following spring dust activities over North China during 1980-2022. The authors demonstrated that the significant impacts of NAO and ENSO on the dust activities over North China is only manifested in the negative phases, and discussed the physical mechanism involved to illustrate why the negative phases of NAO and ENSO show a synergistic effect on the following dust events in North China. The message is conveyed clearly and the topic is interesting. The results of this study provide an insight to further understand the dust activities over North China. The conclusions are substantiated based on composite analyses. If published, this work could serve as a valuable reference for dust weather. However, it needs to be minor revised before accepted this paper for publication in ACP with addressing those comments listed below:

**Response:**

Thanks to the reviewer for the helpful comments and suggestions. We have revised the manuscript seriously and carefully according to the reviewer's comments and suggestions. The point-to-point responses to the comments are listed as follows.

**Specific comments are as follows:**

1. The NAO is a large-scale seesaw in atmospheric mass between the subtropical high and the polar low. It is the dominant mode of atmospheric circulation variability in the North Atlantic sector throughout the year. The definition of the NAO index derived using EOF is commonly employed to depict the variation of NAO. However, the SLP difference between 35°N and 65°N within the Atlantic section is used to define the NAO index. A full comparison of the NAO index is necessary to establish the robustness of result.

**Response:**

Thanks for the comments.

The NAO index (NAOI) employed in the manuscript, is defined as the differences of normalized sea level pressures regionally zonal-averaged over the North Atlantic sector (Li and Wang, 2003). The NAO index captures well large-scale circulation features of the NAO, and is essentially a measure of the intensity of zonal winds across the central North Atlantic between 35°N to 65°N. A systematic comparison of six NAO indices (Rogers, 1984; Barnston and Livezey, 1987; Moses et al., 1987; Hurrell, 1995; Jones et al.,1997; Li and Wang, 2003), shows that the NAOI employed in the manuscript provides a much more faithful and optimal representation of the spatial-temporal variability associated with the NAO, suggesting the NAOI maybe as a suitable choice for describing and monitoring variability of the broad-scale NAO and for diagnosing relationships between the NAO and global climate variations (Li and Wang, 2003).

We also employ the NAOI produce by Hurrell (1995) and Jones (1997), which have been used in many studies (e.g., Wang et al.,2022; Najibi et al., 2023; Parry et al., 2023), for correlation analysis with the NAOI used in this manuscript. A good agreement with correlation coefficients of 0.96 and 0.94 between these two indices and the NAOI used in this manuscript (Figure R1). As well as, using the NAOI provided by Hurrell (1995) and Jones (1997), the asymmetric impact and the synergistic effects with ENSO on dust activities over North China of NAOI still remain (Figures R2-R4). Therefore, the robustness of the results will not be affected by the NAOI verified by

above process. We have added the description into the revised manuscript, as "We also employed the NAOI produce by Hurrell (1995) and Jones (1997), which have been used in many studies (e.g., Wang et al., 2022; Najibi et al., 2023; Parry et al., 2023). A good agreement with correlation coefficients of 0.96 and 0.94 between these two indices and the NAOI defined by Li and Wang (2003)", as shown in Lines 161-164. The relevant reference has been included, and the source of data for the NAOI is included in the "Code and data availability" (Lines 598-602).

[Figure]

**Figure R1**. (a) Normalized time series of NAOI used in the manuscript (black line) and provided by Hurrell (blue line) and Jones (red line) during the previous winter from 1980-2022.

[Figure]

**Figure R2**. Spatial distribution of correlation coefficients between the previous winter NAOI and spring dust content (a). (b-c) As in (a), but for the NAOI produce by Jones and Hurrell, respectively. (d-f) As in (a-c), but for the partial correlation after removing the effect of ENSO. The green box represents North China. Thick and fine stippled areas are statistically significant at the 0.05 and 0.1 level, respectively. The green lines represent the Yellow River (northern one) and the Yangtze River (southern one), respectively.

[Figure]

**Figure R3**. Spatial distribution of correlation coefficients between (a) positive and (b) negative NAOI values and dust content. (c-d) and (e-f), As in (a-b), but for the NAOI produce by Jones and Hurrell, respectively. The green box represents North China. Thick and fine stippled areas are statistically significant at the 0.05 and 0.1 level, respectively. The green lines represent the Yellow River (northern one) and the Yangtze River (southern one), respectively.

[Figure]

**Figure R4**. Spring dust content anomalies over North China under different scenarios. The red/blue bar represents the solo NAO/ENSO negative years, the white bar represents the NAO/ENSO negative years, and green bar represents concurrent negative phases of NAO and ENSO (unit: mg·m$^{-2}$) (a). (b-c) As in (a), but for the NAOI produce by Jones and Hurrell, respectively (unit: mg·m$^{-2}$).

2. The authors focus on the relationship between preceding winter NAO and ENSO and dust weather in late spring. The introduction mentions that "the impacts of winter NAO and ENSO on the climate in China is more pronounced" by citing results from previous work. However, it is unclear whether the cross-seasonal impacts also apply when exploring the relationship between NAO, ENSO and dust weather in North China. Therefore, it would be better to provide some references to explain why we should investigate the impacts of previous winter of NAO and ENSO on spring dust weather.

**Response:**

Thanks for the comments.

The standard deviation of the NAO peaks during December, January, and February. By analyzing the trend of the three-month running average standard deviation, with the maximum during the previous winter. This indicates that the NAO exhibits stronger variability in boreal winter compared to other seasons (Figure R5 a). Similarly, ENSO also shows greater variation during boreal winter (Figure R5 b). Based on these findings, we have chosen to focus on the relationship between NAO, ENSO during the previous winter period, and spring dust activities over North China.

Previous studies have found that preceding NAO and ENSO play important role in impacting the following climate over North China, particular the cross-seasonal impacts (e.g., Zheng et al., 2016; Feng et al., 2019; Sun et al., 2021). We have examined the role of previous autumn, winter and simultaneous spring NAO and ENSO on the spring dust aerosols over North China, and it is found the influences of NAO and ENSO on the spring dust aerosols simultaneously are most significant in the previous winter (Figure R5 c-h). Thus, the role of previous winter NAO and ENSO on the spring dust aerosols over North China are discussed in the present work.

We have added the Fig. R5 (Figure 2 in the revised manuscript) and corresponding descriptions into the revised manuscript (Lines 202-212).

[Figure]

**Figure R5**. The monthly standard deviation of (a) NAOI and (b) Niño3.4 index, respectively. Black line represents three-month running average of standard deviation. (c) Spatial distribution of correlation coefficients between the previous autumn NAOI and spring dust content . (d) As in (c), but with Niño3.4 index. (e-f) and (g-h), as in (c-d), but for the correlations with previous winter and simultaneous spring NAOI and Niño3.4 index, respectively. The green box represents North China. Thick and fine stippled areas are statistically significant at the 0.05 and 0.1 level, respectively. The green lines in (c-h) represent the Yellow River (northern one) and the Yangtze River (southern one), respectively.

3.  In the paper, the authors primarily discuss the effect of NAO and ENSO negative phases on the dust activities over North China. However, given the various phases combinations between these two factors, a more detailed explanation as to why only the negative-negative combinations are considered.

**Response:**

Thanks for the comments.

We have examined the role of different phases of NAO and ENSO on the dust aerosols over North China, and it is found the influences of the positive phases of NAO and ENSO on the dust aerosols are insignificant (Figure 4 in the revised manuscript). "The results indicate that the relationship between NAO/ENSO and dust in North China also exhibits significant asymmetry, i.e., with weaker (stronger) correlations during positive (negative) phases of NAO and ENSO (Figure 4 in the revised manuscript), where significant correlations only appear in the negative phases of NAO and ENSO" (Lines 254-257).

The correlation coefficients between previous winter NAO, ENSO and spring dust aerosol content over North China under different phases are given in Table R1. It further explains that the influence of the negative phases of NAO and ENSO on the dust activities over North China is more significant than when they are in the positive phases. Based on the above discussion, we considered the effect of NAO and ENSO on dust aerosols over North China when they are in the negative phases.

[Figure]

**Figure 4 in the revised manuscript**. Spatial distribution of correlation coefficients between (a) positive and (c) negative NAOI values and dust content. (b) and (d) as in (a) and (b), respectively, but for the Niño3.4 index. The green box represents North China. Thick and fine stippled areas are

statistically significant at the 0.05 and 0.1 level, respectively. The green lines represent the Yellow River (northern one) and the Yangtze River (southern one), respectively.

**Table R1**. Correlation coefficients between the NAOI, ENSO index and regional average dust aerosol content over North China in spring under different phases. * indicates significant at the 0.1 level.

| Scenarios | Correlation coefficients |
|---|---|
| DJF_NAO+ & MAM_DUST | 0.05 |
| DJF_NAO- & MAM_DUST | -0.46* |
| DJF_ENSO+ & MAM_DUST | -0.16 |
| DJF_ENSO- & MAM_DUST | -0.36* |

4. In Figures 1-2, the authors illustrate the relationship between NAO, ENSO and dust weather over North China through the spatial distribution of correlation coefficients, and that the relationship is only manifested when NAO and ENSO are in negative phases. A quantitative analysis is needed to further establish the robustness of the result.

**Response:**

Thanks for the comments.

We have adopted the reviewer's comment and added the quantitative analysis of NAO, ENSO on the dust aerosol content over North China. And we have added the description into the revised manuscript, as "Meanwhile, the NAOI/Niño3.4 index is significantly correlated with the areal averaged spring dust content over North China (SDI), with correlation coefficient of -0.36/-0.35" as shown in Lines 229-230.

As well as, the correlation coefficients of NAO, ENSO during different phases and dust aerosol content over North China in Figure 2 (Figure 5 in the revised manuscript) are described as "Based on the scatter distribution of SDI under different phases of NAO and ENSO, it is noted that the correlation coefficients between NAOI and SDI during the positive and negative phases of NAO are -0.46 and -0.05, respectively, indicating that the significant influence of NAO on the dust in North China mainly

occurs during its negative phase (Figure 5a in the revised manuscript). Similarly, the correlation distribution between the ENSO and SDI also shows that the influence of ENSO is more pronounced during its negative phase (Figure 5b in the revised manuscript)" (Lines 258-263).

[Figure]

**Figure 5 in the revised manuscript**. Scatterplots of the spring dust content in North China against previous winter (a) NAOI and (b) Niño3.4 index. Also shown are lines of best fit for positive and negative NAO/Niño3.4 index values and correlation coefficients (R), slope (slope), * indicates significant at the 0.1 level. (c) Spring dust content over North China during the negative NAO, negative ENSO phases, and concurrent negative phases of NAO and ENSO (unit: mg·m$^{-2}$). White bars represent negative phases of the NAO and ENSO, red and blue bars indicate solo negative NAO and ENSO years, and green bar is the negative NAO and ENSO co-occurring years.

5.  From Fig 3 and Table 1, it is evident that there are two types when NAO and ENSO are in their negative phases: negative phases of the NAO and ENSO, and negative phases of the NAO and ENSO occurring separately (remove the years with concurrent negative phases of NAO and ENSO). Furthermore, the subsequent composite analyses in the study, focus on the cases with negative phases of the NAO and ENSO. The authors should explain why they have made this choice.

**Response:**

Thanks for the comments.

The white bar (labeled as NAO) is the dust content anomalies during the 15 negative NAO years, and red bar is the anomalies for the solo NAO negative years (based on 8 years). The white bar (labeled as ENSO) is the dust content anomalies during the 16 negative ENSO years, and blue bar is the anomalies for the solo ENSO negative years (based on 9 years) (Figure R6).

The above results show that no matter what kind of NAO and ENSO negative phase occurs, the increase in dust aerosol content over North China can be observed. The samples in the case of NAO negative phase are 8 and 15, respectively, and it is of 9 and 16 in the case of ENSO. In order to not only retain the characteristics of the negative phases of NAO and ENSO, but also make our results statistically characteristic, we selected the case of enough samples of the negative phases of NAO and ENSO to consider (Table R2).

[Figure]

**Figure R6**. (a) Spring dust content anomalies over North China under different scenarios. The red/blue bar represents the solo NAO /ENSO negative years, the white bar represents the NAO /ENSO negative years (unit: mg·m$^{-2}$).

**Table R2**. The events of NAO and ENSO.

| Scenarios | Years | Numbers |
|---|---|---|
| NAO$^-$ | 1980,1982,1985,1986,1987,1996,1998,2001, 2003,2004,2006,2010,2011,2013,2021 | 15 |
| ENSO$^-$ | 1984,1985,1986,1989,1996,1999,2000,2001, 2006,2008,2009,2011,2012,2018,2021,2022 | 16 |

6. In Figs 4 c, f, and i, the variations in the near-surface wind field caused by anomalies of Siberian High, lead to dust emissions from the source areas. However, the depiction of the wind field anomalies appears unclear. It is recommended to modify the Figs to highlight the variations in the wind field.

**Response:**

Thanks for the comments and suggestions.

As for the SLP, significant positive SLP anomalies appear in Eastern Europe and Russia during negative NAO phase, indicative of an intensified Siberian High (SH), which extends southward to the dust source regions upstream of North China (Figure 6c in the revised manuscript). The intensification of the SH is typically accompanied by strong northerlies and dry conditions, favoring the transport of dust, thereby supplying abundant material sources for dust activities in North China. In the negative ENSO phase, although the high-latitude region exhibits a weaker SH signal, similar to the ENSO influence on the circulation pattern in the middle and lower troposphere, more significant circulation anomalies occur over the WNP. This cyclonic circulation anomaly inhibits the northward transport of warm and moist air from the south, leading to poorer precipitation conditions in North China (Figure 6f in the revised manuscript). When both the NAO and ENSO are in their negative phases, the strength and influence extent of the SH are more pronounced compared to that when the NAO sole is in negative phase. Additionally, a cyclonic circulation anomaly persists over the WNP, which is conducive to the occurrence of dust events in North China (Figure 6i in the revised manuscript)" (Lines 336-348).

[Figure]

**Figure 6 in the revised manuscript**. Upper, the composite anomalies of (a) 200 hPa zonal wind (shading, unit: m·s$^{-1}$), (b) 500 hPa geopotential height (shading, unit: gpm) and 850 hPa wind field (arrows, unit: m·s$^{-1}$), (c) sea-level pressure (shading, unit: Pa) and 1000 hPa wind field (arrows, unit: m·s$^{-1}$) during the negative NAO phases. Middle-Lower, as in the upper, but during the negative ENSO phases and co-occurred negative phases of NAO and ENSO, respectively. The green box represents North China. Only wind anomalies statistically significant at the 0.1 level are shown. Thick and fine stippled areas are statistically significant at the 0.05 and 0.1 level, respectively.

7.  In Table 2, the value of correlation coefficients between the previous winter NATI and spring NATI are similar in scenarios of ENSO- phase (when the negative phase of ENSO occurs alone) and NAO- & ENSO- phase (when the negative phases of both NAO and ENSO co-occur). However, if there exists a synergistic effect of NAO and ENSO on the dust weather, the correlation in the scenario where both NAO and ENSO negative phases co-occur should be higher than when the negative phases of NAO and ENSO occur separately. The authors should have provided a more detailed explanation to clarify this point.

**Response:**

Thanks for the important comments.

It is seen that the correlation coefficients between the previous winter NATI and the subsequent spring NATI remain consistent across both scenarios (ENSO- phase,

NAO- & ENSO- phase), we would like to clarify this point into the following considerations.

The impacts of previous winter NAO on the spring dust activities over North China are mainly include, 1) The previous winter NAO would stimulate the anomalous NAT SST pattern; 2) The NAT can last from previous winter to the following spring due to the thermal persistence of the SST; 3) The spring NAT plays significant modulation on the circulation pattern over North China through teleconnection wave trains, which ultimately affects the spring dust activities over North China.

It is seen from Table 2 in the manuscript that although in the case of ENSO- phase and NAO- & ENSO- phase, the correlation coefficients of NATI in the previous winter and spring NATI are similar, both of which are 0.69. However, in the process of stimulating NAT by NAO in the previous winter, the correlations between NAO and NAT are higher during NAO- & ENSO- phase (0.66) than ENSO- phase (0.52). This suggests that the NAO significantly drives the NAT in the case of NAO- & ENSO- phase. The above discussion illustrates the synergistic effect of NAO and ENSO on the dust activities over North China (Lines 449-459).

**Table 2 in the revised manuscript**. Correlation coefficients between the NAOI and NATI in three different categories. * indicates significant at the 0.1 level.

| Scenarios | DJF_NAO & DJF _NATI | DJF_NATI & MAM_NATI |
|---|---|---|
| NAO⁻ phase | 0.41* | 0.51* |
| ENSO⁻ phase | 0.52* | 0.69* |
| NAO⁻ & ENSO⁻ phase | 0.66* | 0.69* |

8. The main mechanism for the impact of the winter NAO on the spring dust is the maintenance of the North Atlantic SST anomalies from winter to spring, consistent with previous findings (Chen et al. 2020; Wu and Chen 2020; Song et al. 2022). Several discussions could be added.

**Response:**

Thanks for the comments.

We have quoted the work into the revised manuscript, as "allowing the previous NAO signal to exert a long-term influence on the subsequent weather and climate in

China (e.g., Chen et al., 2020; Wu and Chen, 2020; Song et al., 2022)", as shown in Lines 408-410.

9. There are lots of clerical errors, i.e.,

Line 17-18, sea surface temperatures (SST) in the North Atlantic

Line 220, with regard to the description of the graphs, there may be some errors that"(b) and (d) As in (a) and (b) "-> "(c) and (d) As in (a) and (b)".

The authors should carefully check the whole manuscript.

**Response:**

Thanks to the reviewer for the comments. We have checked the whole manuscript and revised the errors.

**Reference**

Barnston, A. G., and Livezey, R. E.: Classification, seasonality and persistence of low-frequency atmospheric circulation patterns. Mon. Wea. Rev., 115, 1083–1126, https://doi.org/10.1175/1520-0493(1987)115<1083:CSAPOL>2.0.CO;2, 1987.

Chen et al. 2020: Strengthened connection between springtime North Atlantic Oscillation and North Atlantic tripole SST pattern since the late-1980s. Journal of Climate, 35(5), 2007-2022.

Feng, J., Li, J. P., Liao, H., and Zhu, J. L.: Simulated coordinated impacts of the previous autumn North Atlantic Oscillation (NAO) and winter El Niño on winter aerosol concentrations over eastern China, Atmos. Chem. Phys., 19, 10787–10800, https://doi.org/10.5194/acp-19-10787-2019, 2019.

Hurrell, J. W.: Decadal Trends in the North Atlantic Oscillation: Regional Temperatures and Precipitation, Science, 269, 676–679, https://doi.org/10.1126/science.269.5224.676, 1995.

Jones, P. D., Jonsson, T., and Wheeler, D.: Extension to the North AtlanticOscillation using early instrumental pressure observations from Gibraltar and South-West Iceland. Int. J. Climatol., 17, 1433–1450, https://doi.org/10.1002/(SICI)1097-0088(19971115)17:13<1433::AID-JOC203>3.0.CO;2-P, 1997.

Li, J. P., and Wang, J. X. L.: A new North Atlantic Oscillation index and its variability, Adv. Atmos. Sci., 20, 661–676, https://doi.org/10.1007/BF02915394, 2003.

Moses, T., Kiladis, G. N., Diaz, H. F., and Barry, R. G.: Characteristics and frequency of reversals in mean sea level pressure in the North Atlantic sector and their relationship to long-term temperature trends. Int. J. Climatol., 7, 13–30, https://doi.org/10.1002/joc.3370070104, 1987.

Najibi, N., Devineni, N., and Lall, U.: Compound Continental Risk of Multiple Extreme Floods in the United States. Geophys. Res. Lett., 50, e2023GL105297. https://doi.org/10.1029/2023GL105297, 2023.

Parry, S., Lavers, D., Wilby, R., Prudhomme, C., Wood, P., Murphy, C., and OConnor, P.: Abrupt drought termination in the British–Irish Isles driven by high atmospheric vapour transport, Environ. Res. Lett., 18, 104050, https://doi.org/10.1088/1748-9326/acf145, 2023.

Rogers, J. C.: The association between the North Atlantic Oscillation and the Southern Oscillation in the Northern Hemisphere. Mon. Wea. Rev., 112, 1999-2015, https://doi.org/10.1175/1520-0493(1984)112<1999:TABTNA>2.0.CO;2, 1984.

Song, L.-Y., et al, 2022: Distinct evolutions of haze pollution from winter to following spring over the North China Plain: Role of the North Atlantic sea surface temperature anomalies. Atmos. Chem. Phys., 22, 1669–1688.

Sun, L. Y., Yang, X. Q., Tao, L. F., Fang, J. B., and Sun, X. G.: Changing Impact of ENSO Events on the Following Summer Rainfall in Eastern China since the 1950s, J. Climate, 34, 8105–8123, https://doi.org/10.1175/JCLI-D-21-0018.1, 2021.

Wu and Chen, 2020: What leads to persisting surface air temperature anomalies from winter to following spring over the mid-high latitude Eurasia?. Journal of Climate, 33, 5861-5883.

Wang, L., and Ting, M. F.: Stratosphere-Troposphere Coupling Leading to Extended Seasonal Predictability of Summer North Atlantic Oscillation and Boreal Climate, Geophys. Res. Lett., 49, e2021GL096362. https://doi.org/10.1029/2021GL096362, 2022.

Zheng, F., Li, J. P., Li, Y. J., Zhao, S., and Deng, D. F.: Influence of the Summer NAO on the Spring-NAO-Based Predictability of the East Asian Summer Monsoon, J. Appl. Meteorol. Clim., 55, 1459–1476, https://doi.org/10.1175/JAMC-D-15-0199.1, 2016.

**Response to Comments of Reviewer 2**

**Manuscript number**: egusphere-2024-955

**Author(s)**: Falei Xu, Shuang Wang, Yan Li, and Juan Feng

**Title**: Synergistic effects of previous winter NAO and ENSO on the spring dust activities in North China

**General comments:**

Using multi-reanalysis datasets, the authors investigated the effects of the previous winter NAO and ENSO on the spring dust aerosols over North China. The pronounced influence of NAO and ENSO on dust aerosols was predominantly observed during their negative phases. Furthermore, this analysis examined meteorological conditions, atmospheric dynamics, and wave energy transport, elucidating the synergistic impacts of these negative phases on subsequent dust activities. The findings enhance our understanding of the formation mechanisms of dust events in North China. I recommend that this manuscript be accepted after minor revisions, as this study fits well within the scope of Atmospheric Chemistry and Physics.

**Response:**

Thanks to the reviewer for the helpful comments and suggestions. We have revised the manuscript seriously and carefully according to the reviewer's comments and suggestions. The point-to-point responses to the comments are listed as follows.

**Specific comments are as follows:**

1. The study primarily utilizes MERRA-2 reanalysis data for analyzing dust activities over North China. It is essential to assess whether the MERRA-2 data accurately captures dust activities in North China. Please provide further details on the reliability of the reanalysis.

**Response:**

Thanks for the comments.

Previous studies have demonstrated the accuracy and applicability of MERRA-2 reanalysis data for studying the evolution of dust events in Asia. It is reported that the result based on MERRA-2 are similar to those obtained from MODIS, OMPS, CALIPSO, and Himawari-8 data (Kang et al., 2016; Wang et al., 2021).

Additionally, we further employ the datasets from the China National Meteorological Centre from 1980-2018, which include observations of floating dust, blowing dust, and dust storms, to validate the reliability of MERRA-2 reanalysis data. The frequency of dust activities recorded at these stations has been converted into a Dust Index (DI) (Wang et al., 2008; Equations 1), effectively representing the content of dust aerosols.

$$DI = 9 \times DS + 3 \times BD + 1 \times FD \tag{1}$$

Where DS, BD, and FD represent the frequency of dust storms, blowing dust, and floating dust, respectively. Additionally, DI denotes the content of dust aerosols at each station.

We found that the variations of the DI and MERRA-2 dust aerosols content during the four seasons all show similar spatial characteristics (Figure R1). Especially for the dust source in northwest China and the spring dust aerosols over North China, the spatial distribution characteristics are relatively consistent. The above results indicate that the MERRA-2 aerosol reanalysis data can capture the spatiotemporal characteristics of dust aerosol content in China, which is applicable for us to understand the variations in dust aerosol content in China.

We have added the Fig R1 (Figure 1 in the revised manuscript) and corresponding descriptions to the revised manuscript (Lines 126-140).

[Figure]

**Figure R1**. (a-d) Spatial distribution of seasonal mean DI based on station data, (e-h) as in (a-d), but for dust column mass density based on MERRA-2 (units: mg·m⁻²). The green box in (a) and (e) represents North China. The green lines represent the Yellow River (northern one) and the Yangtze River (southern one), respectively.

2. The preceding role of NAO and ENSO on the spring dusty weather over North China is investigated, and it is of interest why the preceding role is focused. And whether their simultaneous role in the dust content is significant or not.

**Response:**

Thanks for the comments.

The standard deviation of the NAO peaks during December, January, and February. By analyzing the trend of the three-month running average standard deviation, with the maximum during the previous winter. This indicates that the NAO exhibits stronger variability in boreal winter compared to other seasons (Figure R2 a). Similarly, ENSO also shows greater variation during boreal winter (Figure R2 b). Based on these findings,

we have chosen to focus on the relationship between NAO, ENSO during the previous winter period, and spring dust activities over North China.

Previous studies have found that preceding NAO and ENSO play important role in impacting the following climate over North China, particular the cross-seasonal impacts (e.g., Zheng et al., 2016; Feng et al., 2019; Sun et al., 2021). We have examined the role of previous autumn, winter and simultaneous spring NAO and ENSO on the spring dust aerosols over North China, and it is found the influences of NAO and ENSO on the spring dust aerosols simultaneously are most significant in the previous winter (Figure R2 c-h). Thus, the role of previous winter NAO and ENSO on the spring dust aerosols over North China are discussed in the present work.

We have added the Fig R2 (Figure 2 in the revised manuscript) and corresponding descriptions to the revised manuscript (Lines 202-212).

[Figure]

**Figure R2**. The monthly standard deviation of (a) NAOI and (b) Niño3.4 index, respectively. Black line represents three-month running average of standard deviation. (c) Spatial distribution of correlation coefficients between the previous autumn NAOI and spring dust content . (d) As in (c), but with Niño3.4 index. (e-f) and (g-h), as in (c-d), but for the correlations with previous winter and simultaneous spring NAOI and Niño3.4 index, respectively. The green box represents North China. Thick and fine stippled areas are statistically significant at the 0.05 and 0.1 level, respectively. The green lines in (c-h) represent the Yellow River (northern one) and the Yangtze River (southern one), respectively.

3. Dust aerosols are important components of atmospheric aerosols, alongside other constituents such as sulfates, nitrogen oxides, black carbon, and so on. Why did the authors choose the dust aerosols in North China as the research objects to be discussed and studied? Please explain and justify this point.

**Response:**

Thanks for the comments.

The dust aerosols are characterized by their significant role within the broader category of atmospheric aerosols. Dust aerosols originate from the mechanical breakdown of rocks and soil into fine particles, which are subsequently transported by the wind (Wang et al., 2018). They are noteworthy for their ability to influence climate systems by affecting solar radiation and cloud formation (e.g., Sokolik and Toon, 1996; Sassen et al., 2003; Zhang et al., 2019). Dust aerosols can pose a formidable threat to socio-economic development, natural ecological environment, as well as human health and safety (e.g., Zhao et al., 2020; Yin et al., 2021; Li et al., 2023). The study of dust aerosols is essential, because understanding their properties and dynamics helps us to better predict weather patterns, assess climate change impacts, and implement effective environmental and public health policies.

4. Line 36-37, "The Gobi Desert in East Asia, especially for the Mongolian Plateau and North China, is a major source of dust". Whether the author is trying to express the meaning of Northern China here, Northern China and North China are two different meanings, please confirm and revise.

**Response:**

Thanks for the comments.

The North China, as a crucial center of politics, economy, and population, refers to the extent of 30°-40°N, 105°-120°E. The northern China (north of 30°N) refers to a broader extent, which encompasses North China, northeast China, Hexi Corridor, and northwest China.

5. Line 77-78, the authors mentioned that "NAO and ENSO often co-occur and have complex interactions". As well as by citing previous work, the facts of a possible relationship between the two factors are enumerated. However, the authors have not thoroughly explored their relationship. It is suggested that further details be provided to enhance the understanding.

**Response:**

Thanks for the comments.

The relationship between the NAO and the ENSO remains unclear. Statistical analyses largely indicate no significant linear association between them. For instance, a correlation analysis of the NAO and ENSO indices during period 1950-2000 shows that their correlation coefficient is only 0.09, suggesting a weak linear correlation (Wang, 2002). Additionally, a significant correlation exists between the La Niña events in autumn and the positive phase of the NAO. However, this is not the case during El Niño events (Pozo-Vazquez et al., 2005).

Recent researchers have further detected the relationship between the NAO and the ENSO, particularly following the identification of two distinct types of ENSO events: the Eastern Pacific (EP) and Central Pacific (CP) El Niño events. It is suggested that the EP El Niño can transmit the Pacific signal to the North Atlantic through a subtropical bridge mechanism, potentially triggering a negative phase in the NAO. However, this relationship is not notably significant (Graf and Zanchettin, 2012). Moreover, the atmospheric circulation in the North Atlantic region reacts differently to the two types of La Niña events. During an EP La Niña, when the North Atlantic jet is weakened, the NAO tends to be in a negative phase. Conversely, CP La Niña would

strengthen the North Atlantic jet, and the NAO is more likely to exhibit a positive phase (Zhang et al., 2015).

The above discussion suggests that there may be a nonlinear link between NAO and ENSO, and the relationship between them is still inconclusive and requires further study. Therefore, this paper analyzes the synergistic effect of NAO and ENSO on dust activities over North China.

6.  In Figure 3 (a), it is notable that there is a point during the negative phase of the NAO that deviates from the majority of the points, potentially qualifying it as an outlier. If this point is removed from the sequence, it is important to verify whether the relationship between the NAO and dust aerosol content remains robust.

**Response:**

Thanks for the important comments.

It is important to consider whether the influence of the negative phase of the NAO on dust activities in North China persists after removing the outlier. When the outlier is excluded, we observe a reduction in the correlation between the two factors, yet a significant correlation remains and passes the 0.2 statistical significance test (Figure R3). This suggests that the impact of the NAO on dust activities in North China is robust during its negative phase.

[Figure]

**Figure R3**. Scatterplots of the spring dust content in North China against previous winter (a) NAOI and (b) NAOI (Remove Outlier). Also shown are lines of best fit for positive and negative NAO/Niño3.4 index values and correlation coefficients (R), slope (slope), * indicates significant at the 0.2 level.

7.  In Figure 5, by describing the precipitation and humidity fields under different scenarios, the authors illustrate the synergistic effect of NAO and ENSO on dust

activities in North China. However, the large values of the variables in the graphs do not seem to be well highlighted. Consider modifying the color-coded intervals to enhance the reader's understanding of the section.

**Response:**

Thanks for the comments and suggestions.

We have revised the figures to highlight large values of the precipitation and humidity fields under different scenarios. "When the NAO is in its negative phase, humidity in the spring dust source regions and North China is generally reduced, particularly in areas near the dust source regions, indicating that these areas are conducive to dust transport and prone to causing dust activities in North China (Figure 7a in the revised manuscript). As for the precipitation, there is more spring precipitation in the northwest region of China, while precipitation in the Mongolia and the North China is relatively less (Figure 7b in the revised manuscript). In the negative ENSO phase, the variation in humidity is similar to that during the negative NAO phase, but with a greater amplitude (Figure 7c in the revised manuscript), indicating that ENSO has a stronger impact on the humidity conditions in North China. Moreover, the precipitation shows a significant decrease over Mongolia and North China, which is highly conducive to dust activities (Figure 7d in the revised manuscript). When both the NAO and ENSO are in the negative phases, the humidity anomalies in the dust source regions and North China are more intense than the individual factor (Figure 7e in the revised manuscript). The variation in precipitation is similar to those in humidity, the reduction in precipitation in the dust source regions and North China exceeds the sole role (Figure 7f in the revised manuscript)" (Lines 372-385).

[Figure]

**Figure 7 in the revised manuscript**. Upper, composite percentage anomalies of (a) special humidity and (b) precipitation during negative NAO phases. Middle-Lower, as in the upper, but during negative ENSO phases and concurrent negative phases of NAO and ENSO, respectively. The green box represents North China. Thick and fine stippled areas are statistically significant at the 0.05 and 0.1 level, respectively. The green lines represent the Yellow River (northern one) and the Yangtze River (southern one), respectively.

**Reference**

Feng, J., Li, J. P., Liao, H., and Zhu, J. L.: Simulated coordinated impacts of the previous autumn North Atlantic Oscillation (NAO) and winter El Niño on winter aerosol concentrations over eastern China, Atmos. Chem. Phys., 19, 10787–10800, https://doi.org/10.5194/acp-19-10787-2019, 2019.

Graf, H.-F. and Zanchettin, D.: Central Pacific El Niño, the "subtropical bridge," and Eurasian climate, J. Geophys. Res.-Atmos., 117, https://doi.org/10.1029/2011JD016493, 2012.

Kang, L. T., Huang, J. P., Chen, S. Y., and Wang, X.: Long-term trends of dust events over Tibetan Plateau during 1961-2010, Atmos. Environ., 125, 188-198, https://doi.org/10.1016/j.atmosenv.2015.10.085, 2016.

Li, Y., Xu, F. L, Feng, J., Du, M. Y., Song, W. J., Li, C., and Zhao, W. J.: Influence of the previous North Atlantic Oscillation (NAO) on the spring dust aerosols over North China, Atmos. Chem. Phys., 23, 6021–6042, https://doi.org/10.5194/acp-23-6021-2023, 2023.

Pozo-Vazquez, D., Gamiz-Fortis, S. R., Tovar-Pescador, J., Esteban-Parra, M. J., and Castro-Diez, Y.: North Atlantic winter SLP anomalies based on the autumn ENSO state, J. Clim., 18, 97–103, 2005.

Sassen, K., DeMott, P. J., Prospero, J. M., and Poellot, M. R.: Saharan dust storms and indirect aerosol effects on clouds: CRYSTAL-FACE results, Geophys. Res. Lett., 30, 1633, https://doi.org/10.1029/2003GL017371, 2003.

Sokolik, I. N. and Toon, O. B.: Direct radiative forcing by anthropogenic airborne mineral aerosols, Nature, 381, 681–683, https://doi.org/10.1038/381681a0, 1996.

Sun, L. Y., Yang, X. Q., Tao, L. F., Fang, J. B., and Sun, X. G.: Changing Impact of ENSO Events on the Following Summer Rainfall in Eastern China since the 1950s, J. Climate, 34, 8105–8123, https://doi.org/10.1175/JCLI-D-21-0018.1, 2021.

Wang C. Z.: Atlantic Climate Variability and Its Associated Atmospheric Circulation Cells, J. Climate, 15, 1516-1536, https://doi.org/10.1175/1520-0442(2002)015<1516:ACVAIA>2.0.CO;2, 2002.

Wang, T. H., Tang, J. Y., Sun, M. X., Liu, X. W., Huang, Y. X., Huang, J. P., Han, Y., Cheng, Y. F., Huang, Z. W., and Li, J. M.: Identifying a transport mechanism of dust aerosols over South Asia to the Tibetan Plateau: A case study, Sci. Total Environ., 758, 11, https://doi.org/10.1016/j.scitotenv.2020.143714, 2021.

Wang, X., Huang, J. P., Ji, M. X., and Higuchi, K.: Variability of East Asia dust events and their long-term trend, Atmos. Environ., 42, 3156-3165, https://doi.org/10.1016/j.atmosenv.2007.07.046, 2008.

Wang, X., Liu, J., Che, H. Z., Ji, F., and Liu, J. J.: Spatial and temporal evolution of natural and anthropogenic dust events over northern China, Sci. Rep., 8, 2141, https://doi.org/10.1038/s41598018-20382-5, 2018.

Yin, Z. C., Wan, Y., Zhang, Y. J., and Wang, H. J.: Why super sandstorm 2021 in North China? Natl. Sci. Rev., 9, nwab165, https://doi.org/10.1093/nsr/nwab165, 2021.

Zhang, C. X., Liu, C., Hu, Q. H., Cai, Z. N., Su, W. J., Xia, C. Z., Zhu, Y. Z., Wang, S. W., and Liu, J. G.: Satellite UV-Vis spectroscopy: implications for air quality

trends and their driving forces in China during 2005–2017, Light-Sci. Appl., 8, 100, https://doi.org/10.1038/s41377-019-0210-6, 2019.

Zhang, W. J., Wang, L., Xiang, B. Q., Qi, L., and He, J. H.: Impacts of two types of La Nina on the NAO during boreal winter, Clim. Dyn., 44, 1351-1366, https://doi.org/10.1007/s00382-014-2155-z, 2015.

Zhao, C. F., Yang, Y. K., Fan, H., Huang, J. P., Fu, Y. F., Zhang, X. Y., Kang, S. C., Cong, Z. Y., Letu, H., and Menenti, M.: Aerosol characteristics and impacts on weather and climate over the Tibetan Plateau, Natl. Sci.Rev., 7, 492–495, https://doi.org/10.1093/nsr/nwz184, 2020.

Zheng, F., Li, J. P., Li, Y. J., Zhao, S., and Deng, D. F.: Influence of the Summer NAO on the Spring-NAO-Based Predictability of the East Asian Summer Monsoon, J. Appl. Meteorol. Clim., 55, 1459–1476, https://doi.org/10.1175/JAMC-D-15-0199.1, 2016.

---

## Author Response (AR3)

**Response to Comments of Editor**

**Manuscript number**: egusphere-2024-955

**Author(s)**: Falei Xu, Shuang Wang, Yan Li, and Juan Feng

**Title**: Synergistic effects of previous winter NAO and ENSO on the spring dust activities in North China

**General comments:**

1. Setting the threshold for statistical significance to p=0.10 resulted in a reduction of overall significance of your results. Therefore, 1) the discussion of some figures should be adjusted to the new results; 2) the overall low statistical significance of the results should be highlighted and discussed in the conclusions.

**Response:**

Thanks for the suggestions.

We have revised the relevant discussions to adjust the new results throughout the manuscript.

We have added the corresponding descriptions, as shown "The present study focuses on the period 1979-2022, due to the longevity of the MERRA-2 dust content dataset. There are only 7 co-occurrence years of negative NAO and ENSO, which take up to 17% of the whole study period. It is noted that the co-occurrence events are not as many as either the negative NAO or ENSO, thus a significance level of 0.1 is displayed. It is worthy to examine their joint impacts by employing longer datasets or models outputs, to further explore their synergistic effects and any possible variations in their modulations" in Lines 537-543.

2. English still needs to be improved, there are still many typos and mistakes, odd expressions and some sentences don't read fluently. I recommend a further deep revision of the language.

**Response:**

Thanks for the suggestions.

We have done our best to revise the manuscript. Additionally, we have conducted a thorough language check to enhance overall clarity and coherence.

**Specific comments to the manuscript:**

1.  L21: WNP is not used elsewhere in the abstract, please remove it.

**Response:**

Yes, done.

2.  L52: westerly belt, do you mean the mid-latitude westerly regime.

**Response:**

Thanks for the comments.

We have revised it to "mid-latitude westerly regime" (Line 49).

3.  Figure 1: how do you get a spatialised map from station data? Did you apply any spatialisation technique? Please clarify. Also specify the unit for DI.

Thanks for the comments and suggestions.

In order to better compare the DI with the reanalysis, we first interpolate the site data into grid points by Cressman (1959), and then obtain the gridded DI. We have added corresponding descriptions into the revised manuscript (Lines 132-134).

$$DI = 9 \times DS + 3 \times BD + 1 \times FD \qquad (1)$$

It is worth noting that the value of 1 represents the normalized mass weight of dust content for each FD, while 3 and 9 represent the relative mass weight of dust content for BD and DS, respectively (Equation 1) (Wang et al., 2008). Therefore, DI is an index used to indicate the dust content which does not have unit. We have included the corresponding revision in the revised manuscript (Lines 129-132).

4.  L137: "overall consistent".

**Response:**

Yes, done.

5. L149: please clarify in the text whether winter 2022 is defined from December 2021 or December 2022.

**Response:**

Thanks for the suggestions.

The winter are during 1979-2021, and the spring are during 1980-2022, to highlight the preceding impacts of previous winter NAO and ENSO on the following spring dust activities over North China. The winter 2022 is not considered in the manuscript due to the longevity of the dataset.

We have added corresponding descriptions, as shown "The winter is defined as the average of December-February (December-January-February, DJF), with the winter 1979 (2021) corresponding to the average of December in 1979 (2021), January and February in 1980 (2022). The spring seasonal mean is the average of March, April, and May. Thus, the previous winter is from 1979 to 2021, and the following spring is from 1980 to 2022" in Lines 143-146.

6. L161: NAOI from Hurrell and Jones have been used to validate the NAOI by Li and Wang.

**Response:**

Yes, done.

7. L166: the justification for focusing on previous winter is given in Sec. 3.1. Here you can just tell how seasonal means are computed.

**Response:**

Thanks for the suggestions.

We have included the seasonal means in Sec. 2.1. The winter is defined as the average of December-February (December-January-February, DJF), with the winter 1979 (2021) corresponding to the average of December in 1979 (2021), January and February in 1980 (2022). The spring seasonal mean is the average of March, April, and

May. Thus, the previous winter is from 1979 to 2021, and the following spring is from 1980 to 2022.

8. L195: please revise the definitions: p should be pressure and U should be U=(U, V, 0) (you need a 3D vector wind for a 3D formulation). What is the difference between f0 and f? What is N? what is z?

**Response:**

Thanks for the suggestions and comments.

We have revised the definitions, as shown "In the expression, $p$, $\varphi$, $\lambda$, $f_0$, and $a$ represent the atmospheric pressure, latitude, longitude, Coriolis parameter, and Earth's radius, respectively. $\psi' = \Phi'/f_0$ (where $\Phi$ represents the geopotential height) denotes the disturbance of the quasi-geostrophic stream function relative to the climatology. $N$ is buoyancy frequency, $z = -H ln(p)$ with $H$ being a constant scale height ($H$=8 km). The basic flow field $\boldsymbol{U} = (U, V, Z)$ (where $Z$ represents the selected level) denotes the climatic field, where $U$ and $V$ indicate the zonal and meridional velocities, respectively" in Lines 189-194.

9. L202: NAO std actually peaks in February.

**Response:**

Thanks for the comments.

We have revised it, as shown "The NAO shows the strongest variability during the winter months, with the maximum standard deviation in February" in Lines 197-198.

10. Figure 2: I'm wondering why, being the NAOI standardised, the std in panel (a) shows such high values. Shouldn't be around 1 on an annual basis? As shown by the ENSO index in panel (b).

**Response:**

Thanks for the comments.

In Figure 2, it is the raw series instead of the standardized series to calculate the standard deviation of NAOI and ENSO index.

11. L230: are the correlation coefficients significant?

**Response:**

Thanks for the comments.

We have revised it, as shown "with correlation coefficients of -0.36/-0.35 statistically significant at the 0.1 level" in Lines 217-218.

12. Figure 3: panels (a) and (b) are the same as in Fig. 2, please remove them and modify caption and text accordingly.

**Response:**

Thanks for the suggestions.

We have revised the figure and caption (Figure 3 in the revised manuscript, also seeing in Figure R1). And we have modified the text accordingly in the revised manuscript.

[Figure]

**Figure R1**. (a) Spatial distribution of partial correlation coefficients between the previous winter NAOI and spring dust content after removing the effect of ENSO. (b) As in (a), but after removing the effect of NAO. The green box represents North China. Thick and fine stippled areas are statistically significant at the 0.05 and 0.1 level, respectively. The green lines represent the Yellow River (northern one) and the Yangtze River (southern one), respectively.

13. L260: please provide information about the significance of correlations.

**Response:**

Yes, done.

14. L262: what do you mean by correlation distribution? Is it not just correlation?

**Response:**

Thanks for the comments.

We have revised it to "the correlation coefficients between NAOI and SDI" (Line 244).

15. L263: please provide information about correlation coefficients and significance.

**Response:**

Yes, done.

16. L264: ENSO in brackets is confusing, please rephase in a more readable way.

**Response:**

Thanks for the suggestions.

We have revised it, as shown "These results demonstrate that the impacts of the previous winter NAO and ENSO on the SDI exhibit asymmetrical characteristics, with significant effects primarily manifested during their negative phases" in Lines 250-252.

17. Figure 5: thanks for improving the description of the figure, which is now much clearer. I also think that error bars are no longer necessary (I apologise for changing my mind on that, it's just that I couldn't understand the plot). I rather suggest that you provide information about the statistical significance of the SDI in the different cases. Specifically: NAO vs NAO-; ENSO vs ENSO-; NAO vs NAO&ENSO; ENSO vs NAO&ENSO.

**Response:**

Thanks for the suggestions.

The statistical significance of the SDI in different cases is shown in Figure R2. It is noted that the SDI anomalies are statistical insignificant when negative NAO or ENSO occurs alone. This suggests that the overlapped negative NAO and ENSO events show a synergistic effect on dust activities in North China.

[Figure]

**Figure R2**. Spring dust content over North China in the different cases. * indicates statistically significant at the 0.1 level.

18. L309: zonal wind is stronger in NAO&ENSO case wrt the NAO- case, but not so clear wrt the ENSO- case. You should provide robust evidence, or mitigate this claim.

**Response:**

  Thanks for the suggestions.

  We have made the corresponding revisions in the revised manuscript (Lines 287-298).

19. Figure 7: after the change in the significance level, this figure is now problematic. The NAO- case show no significant anomalies, while ENSO- and NAO&ENSO cases only show significant anomalies north west of North China, and no significant anomalies in North China. You should modify the discussion of this figure in the light of the new results.

**Response:**

  We have modified the discussion of this figure in the light of the new results, as shown "During the negative NAO phase, humidity and precipitation slightly decrease in northern northwest China, impacting dust lifting and transport in the dust source

regions (Figures 7a-b). In the negative ENSO phase, the variations in humidity and precipitation are similar to that as in the negative NAO, but with greater amplitude (Figures 7c-d). When both the NAO and ENSO are in their negative phases, the humidity and precipitation anomalies in the dust source regions are more intense than those caused by the individual factors (Figure 7e-h)" in Lines 353-359.

20. L415: Figures 8bc don't show the SDI-NATI correlation, but the SDI-SST correlation.

**Response:**

Thanks for the suggestions.

We have revised it, as shown "The correlation analysis between the high and low years of SDI and SST reveals a pronounced difference, indicating an asymmetric correlation (Figures 8b-c)" in Lines 385-387.

21. L416: where do we see the significant relationship between SDI and NATI?

**Response:**

Thanks for the comments.

We have revised the description, as shown "Specifically, the significant relationship between SDI and NATI only exists in the positive SDI years, with a significant correlation coefficient of -0.47, implying that the occurrence of NATI would associate with more dust activities over North China" in Lines 387-390.

22. L429: please provide significance for the correlations.

**Response:**

Yes, done.

23. Figure 10g-i: the first impression here is that the correlation pattern is not stronger than in the NAO- and ENSO- cases. You should provide quantitative evidence, or mitigate the claim at L481-485.

**Response:**

Thanks for the suggestions.

We have made the corresponding revisions in the text to mitigate this claim, as shown "Notably, when both the NAO and ENSO are in their negative phases, the correlation patterns of the teleconnection structure are similar, however the anomalies over North China is enhanced, showing significant anomalies in the vorticity field (Figures 10g-i), confirming their synergistic effects on the circulation processes affecting dust activities in North China" in Lines 446-450.

24. Figures 10, 11 and 12: you should explain (in the caption) why regressions fields are multiplied by -1.

**Response:**

Thanks for the suggestions.

We have revised the caption in Figure 10, 11 and 12, as shown "Regression fields have multiplied by -1 (to facilitate a direct comparison between the NAO&ENSO associated circulation anomalies and the climatology)".

25. Figures 10 and 12: please display North China.

**Response:**

Yes, done

26. Section 4: please discuss in this section the limitations associated with the overall low statistical significance of your results.

**Response:**

Thanks for the suggestions.

We have added the discussions in Section 4, as shown "The present study focuses on the period 1979-2022, due to the longevity of the MERRA-2 dust content dataset. There are only 7 co-occurrence years of negative NAO and ENSO, which take up to 17% of the whole study period. It is noted that the co-occurrence events are not as many as either the negative NAO or ENSO, thus a significance level of 0.1 is displayed. It is worthy to examine their joint impacts by employing longer datasets or models outputs,

to further explore their synergistic effects and any possible variations in their modulations" in Lines 537-543.

27. L541: please avoid using acronyms that you don't use much in this section, e.g. WNP, SH.

**Response:**

Yes, done.

28. L573: please clarify how "the availability of dataset" affects the study of the interdecadal variability.

**Response:**

Thanks for the comments.

The present work mainly focuses the interannual modulation of NAO and ENSO on the dust activities over North China, however, the NAO and ENSO, as well as dust activities over North China, bear strong interdecadal variations, long-term datasets are needed to further explore their impacts on the dust activities (Lines 533-537).

29. L587: this last sentence could be moved to the discussion of the interdecadal variability above.

**Response:**

Thanks for the suggestions.

We have revised it in the revised manuscript (Lines 546-548).

**Reference**

Wang, X., Huang, J. P., Ji, M. X., and Higuchi, K.: Variability of East Asia dust events and their long-term trend, Atmos. Environ., 42, https://doi.org/10.1016/j.atmosenv.2007.07.046, 2008.

Cressman, G. P.: An operational objective analysis system, Mon. Weather Rev., 87, 367–374, https://doi.org/10.1175/1520-0493(1959)087<0367:AOOAS>2.0.CO;2, 1959.